# Hypergraphs as Weighted Directed Self-Looped Graphs: Spectral Properties, Clustering, Cheeger Inequality

**Zihao Li**                                                                 *zihaoli5@illinois.edu*
*University of Illinois Urbana-Champaign*

**Dongqi Fu**                                                                 *dongqifu@meta.com*
*Meta*

**Hengyu Liu**                                                               *hengyu2@illinois.edu*
*University of Illinois Urbana-Champaign*

**Jingrui He**                                                               *jingrui@illinois.edu*
*University of Illinois Urbana-Champaign*

**Reviewed on OpenReview:** *https://openreview.net/forum?id=xLWhuCXWiM*

## Abstract

Hypergraphs naturally arise when studying group relations and have been widely used in the field of machine learning. To the best of our knowledge, the recently proposed **_e_**_dge-_**_d_**_ependent_ **_v_**_ertex_ **_w_**_eights_ (EDVW) modeling (Chitra & Raphael, 2019) is one of the most generalized modeling methods of hypergraphs, i.e., most existing hypergraph conceptual modeling methods can be generalized as EDVW hypergraphs without information loss. However, the relevant algorithmic developments on EDVW hypergraphs remain nascent: compared to the spectral theories for graphs, *its formulations are incomplete, the spectral clustering algorithms are not well-developed, and the hypergraph Cheeger Inequality is not well-defined*. To this end, deriving a unified random walk-based formulation, we propose our definitions of hypergraph Rayleigh Quotient, NCut, boundary/cut, volume, and conductance, which are consistent with the corresponding definitions on graphs. Then, we prove that the normalized hypergraph Laplacian is associated with the NCut value, which inspires our proposed **HyperClus-G** algorithm for spectral clustering on EDVW hypergraphs. Finally, we prove that HyperClus-G can always find an approximately linearly optimal partitioning in terms of both NCut[1] and conductance[2]. Additionally, we provide extensive experiments to validate our theoretical findings from an empirical perspective.

## 1 Introduction

Higher-order relations are ubiquitous, such as co-authorship (Feng et al., 2018; Yadati et al., 2019; Sun et al., 2021), interactions between multiple proteins or chemicals (Feng et al., 2021; Xia et al., 2022), items that are liked by the same person (Yang & Leskovec, 2015; Wu et al., 2021), and interactions between multiple species in an ecosystem (Grilli et al., 2017; Sanchez-Gorostiaga et al., 2019). Hypergraphs, extended from graphs, with the powerful capacity to model group interactions (i.e., higher-order relations), show extraordinary potential for real-world tasks beyond pair-wise connections. Therefore, hypergraphs have been used widely

---

[1]The NCut of the returned partition $\mathcal{N}$ and the optimal NCut of any partition $\mathcal{N}^*$ satisfy $\mathcal{N} \leq O(\mathcal{N}^*)$.
[2]The conductance of the returned partition $\Phi$ and the optimal conductance $\Phi^*$ satisfy $\Phi \leq O(\Phi^*)$

Table 1: Properties of graph models/formulations. EDVW hypergraphs generalized EIVW hypergraphs by allowing each hyperedge to distribute its vertex weights, bringing better formulation flexibility.

| Modeling/Formulation | undirected graphs | EIVW hypergraphs | EDVW hypergraphs |
|---|---|---|---|
| edge and vertex weights | $\checkmark$ | $\checkmark$ | $\checkmark$ |
| hyperedges | $\times$ | $\checkmark$ | $\checkmark$ |
| edge-dependent vertex weights | $\times$ | $\times$ | $\checkmark$ |

Table 2: A summary of our developed formulations of EDVW hypergraph spectral properties. Based on random walks on EDVW hypergraphs, we further develop the Rayleigh Quotient, NCut, boundary/cut, volume, and conductance, as well as their associations. "-" means not studied in Chitra & Raphael (2019).

| Terminology/Property | Graphs | EDVW Hypergraphs | |
|---|---|---|---|
| | | Chitra & Raphael (2019) | Our work |
| Graph Definition | $\mathcal{G} = (\mathcal{V}, \mathcal{E})$ | $\mathcal{H} = (\mathcal{V}, \mathcal{E}, \omega, \gamma)$. $\omega$ stores edge weight and $\gamma$ stores EDVW. | |
| Random Walk Transition | $P_{u,v} = \frac{\mathbb{1}((u,v)\in\mathcal{E})}{degree(u)}$ | $P_{u,v} = \sum_{e\in E(u)} \frac{\omega(e)}{d(u)}\frac{\gamma_e(v)}{\delta(e)}$, with row sum of 1. | |
| Stationary Distribution | $\phi P = \phi$ and $\sum_u \phi(u) = 1$ | $\phi P = \phi$ and $\sum_u \phi(u) = 1$, diagonal matrix $\Pi(v,v) = \phi(v)$ | |
| Laplacian(s) | $L = D - A$, $L_{sym} = D^{-\frac{1}{2}} L D^{-\frac{1}{2}}$ | $L = \Pi - \frac{\Pi P + P^T \Pi}{2}$ | $L = \Pi - \frac{\Pi P + P^T \Pi}{2}$, $L_{sym} = \Pi^{-\frac{1}{2}} L \Pi^{-\frac{1}{2}}$ |
| Boundary Volume | $|\partial S| = \sum_{u\in\mathcal{S}, v\in\bar{\mathcal{S}}} \mathbb{1}((u,v)\in\mathcal{E})$ | - | $|\partial S| = \sum_{u\in\mathcal{S}, v\in\bar{\mathcal{S}}} \phi(u)P_{u,v}$ |
| Vertex Set Volume | $vol(S) = \sum_{u\in\mathcal{S}} degree(u)$ | - | $vol(S) = \sum_{u\in\mathcal{S}} \phi(u)$ |
| Rayleigh Quotient | $R(x) = \frac{\sum_{(u,v)\in\mathcal{E}} |x(u)-x(v)|^2}{\sum_{u\in\mathcal{V}} |x(u)|^2}$ | - | $R(x) = \frac{\sum_{u,v} |x(u)-x(v)|^2 \phi(u)P_{u,v}}{\sum_u |x(u)|^2 \phi(u)}$ |
| Normalized Cut ($NCut$) | $(\frac{1}{vol(\bar{\mathcal{S}})} + \frac{1}{vol(\mathcal{S})})|\partial\mathcal{S}|$ | | $NCut(\mathcal{S}, \bar{\mathcal{S}}) = (\frac{1}{vol(\mathcal{S})} + \frac{1}{vol(\bar{\mathcal{S}})})|\partial\mathcal{S}|$ |
| Conductance | $\Phi(\mathcal{S}) = \frac{|\partial\mathcal{S}|}{\min(vol(\mathcal{S}), vol(\bar{\mathcal{S}}))}$ | - | $\Phi(\mathcal{S}) = \frac{|\partial\mathcal{S}|}{\min(vol(\mathcal{S}), vol(\bar{\mathcal{S}}))}$ |
| Algebraic Connections | similar property as our theorem 1 | | $NCut(\mathcal{S}, \bar{\mathcal{S}}) = \frac{1}{2}R(x) = \frac{x^T L x}{x^T \Pi x}$, Theorem 1 |
| Cheeger Inequality | $\frac{\Phi(\mathcal{G})^2}{2} \leq \lambda_{L_{sym}} \leq 2\Phi(\mathcal{G})$ | $\frac{\Phi(\mathcal{H})^2}{2} \leq \lambda_L \leq 2\Phi(\mathcal{H})$ $\times$ | $\frac{\Phi(\mathcal{H})^2}{2} \leq \lambda_{L_{sym}} \leq 2\Phi(\mathcal{H})$, Theorem 3 $\checkmark$ |
| Spectral Clustering | well-developed | - | a new Algorithm 1 for EDVW hypergraphs |

in recommendation systems (Zhu et al., 2016; Liu et al., 2022; Gatta et al., 2023), information retrieval (Huang et al., 2009; Zhu et al., 2015) and link prediction (Huang et al., 2020; Fan et al., 2022).

Hypergraphs modeled by **_edge-dependent vertex weights_** (EDVW) were necessitated in a recent work (Chitra & Raphael, 2019), with a motivating example that in citation networks, each scholar (i.e., node) may contribute differently to each co-authored publication (i.e., hyperedge). The authors show that hypergraphs with **_edge-**_**in**_**dependent vertex weights_** (EIVW) do not actually utilize the higher-order relations for the following two reasons. First, the hypergraph Laplacian matrix proposed by the seminal work (Zhou et al., 2006), which serves as a basis of many follow-up algorithms, is equal to the Laplacian matrix of a closely related graph with only pair-wise relations. In this way, all the linear Laplacian operators utilize only pair-wise relationships between vertices (Agarwal et al., 2006). Second, many hypergraph algorithms (Ma et al., 2018; Li et al., 2018; Carletti et al., 2020) are based on random walks (Tong et al., 2006), but it has been proved that for any EIVW hypergraph, there exists a weighted pair-wise graph on which a random walk is equivalent to that on the original hypergraph (Chitra & Raphael, 2019).

In nature, EDVW hypergraphs are not a special case of hypergraphs, but a more generalized way to model hypergraphs by allowing flexible weight distribution, as shown in Figure 1 [revised as a side figure]. As typical (EIVW) hypergraphs can be reformulated to EDVW modeling, the properties and algorithms of EDVW-formulated hypergraphs can be applied to EIVW hypergraphs.

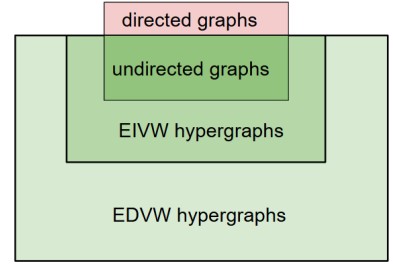

Figure 1: **Undirected graphs $\subset$ EIVW hypergraphs $\subset$ EDVW hypergraphs**. Pairwise edges are naturally hyperedges; each EIVW hypergraph can be reformulated to an EDVW hypergraph by setting each vertex's weight to be the same across hyperedges, yet allowing different vertices to have different weights.

In this paper, we focus on further developing the incomplete yet fundamental spectral theories for EDVW hypergraphs, with a straightforward application on spectral clustering, i.e., $k$-way global partitioning, where typically $k = 2$. To be specific, $k$-way global partitioning aims to partition an entire graph into $k$ clusters, where the vertices in one cluster are densely connected within this cluster while having sparser connections to vertices outside this cluster. On the one hand, although the spectral theories and spectral clustering on graphs have been well studied, converting the hypergraphs to graphs (by connecting each node in a hyperedge) and applying those classic graph methods may ignore the higher-order relations and result in sub-optimal results (Wang & Kleinberg, 2024). On the other hand, despite the advantage of EDVW modeling, directly developing a spectral clustering algorithm on EDVW hypergraphs is still an open question. To this end, for the first time, we propose a provably linearly optimal spectral clustering algorithm on EDVW hypergraphs, together with theoretical analysis concerning the Rayleigh Quotient, Normalized Cut (i.e., NCut), and conductance. In the context of EDVW hypergraphs, we bridge the eigensystem of Laplacian with the NCut value through our proposed Rayleigh Quotient. As discussed previously, the proposed algorithm can be applied to EIVW hypergraphs as well.

## 1.1 Main Results

In this paper, we further develop the spectral hypergraph theory for EDVW hypergraphs, and then study global partitioning on EDVW hypergraphs.

**Theorem 1.** *(Algebraic connections among hypergraph NCut, Rayleigh Quotient, and Laplacian) Given any hypergraph $\mathcal{H}$ with vertex set $\mathcal{V}$ and hyperedge set $\mathcal{E}$ in the EDVW formatting, i.e., $\mathcal{H} = (\mathcal{V}, \mathcal{E}, \omega, \gamma)$ with positive edge weights $\omega(\cdot) > 0$ and non-negative edge-dependent vertex weights $\gamma_e(\cdot)$ for any hyperedge $e \in \mathcal{E}$, we define Normalized Cut $NCut(\cdot)$, Volume of a vertex set $vol(\cdot)$, Rayleigh Quotient $R(\cdot)$, Laplacian $L$, and stationary distribution matrix $\Pi$ as Definition 16, 13, 14, 9, and 7. For any vertex set $\mathcal{S} \subseteq \mathcal{V}$, we define a $|\mathcal{V}|$-dimensional vector $x$ such that*

$$
\begin{aligned}
x(u) &= \sqrt{\frac{vol(\bar{\mathcal{S}})}{vol(\mathcal{S})}}, \ \forall \ u \in \mathcal{S}, \\
x(\bar{u}) &= -\sqrt{\frac{vol(\mathcal{S})}{vol(\bar{\mathcal{S}})}}, \ \forall \ \bar{u} \in \bar{\mathcal{S}}.
\end{aligned}
\tag{1}
$$

*Then,*

$$
NCut(\mathcal{S}, \bar{\mathcal{S}}) = \frac{1}{2} R(x) = \frac{x^T L x}{x^T \Pi x}
\tag{2}
$$

To the best of our knowledge, while prior works Jost & Mulas (2019); Mulas et al. (2022) studied the Rayleigh Quotient under the EIVW model, this is the first work that develops the Rayleigh Quotient for EDVW hypergraphs. Inspired by this Theorem, we develop a spectral clustering algorithm HyperClus-G to optimize the NCut value by relaxing the combinatorial optimization constraint.

**Theorem 2.** *(Hypergraph Spectral Clustering Algorithm) There exists an algorithm for hypergraph spectral clustering that can be applied to EDVW-formatted hypergraphs, and always returns approximately linearly optimal clustering in terms of Normalized Cut and conductance. In other words, approximately, the NCut of the returned partition $\mathcal{N}$ and the optimal NCut of any partition $\mathcal{N}^*$ satisfy $\mathcal{N} \leq O(\mathcal{N}^*)$.*

We name this algorithm as HyperClus-G, whose pseudo code is given in Algorithm 1 with complexity analyzed in Appendix B. Moreover, to extend the hypergraph spectral theory, for the first time, we give complete proof regarding the hypergraph Cheeger Inequality. In the meantime, by proving Theorem 3, the previous result on hypergraph Cheeger Inequality (Theorem 5.1 in (Chitra & Raphael, 2019)) is fixed and upgraded.

**Theorem 3.** *(Hypergraph Cheeger Inequality) [reduced verbosity] Let $\mathcal{H} = (\mathcal{V}, \mathcal{E}, \omega, \gamma)$ be a hypergraph in the EDVW formatting. Define $\Phi(\mathcal{H}) = \min_{\mathcal{S} \subseteq \mathcal{V}} \Phi(\mathcal{S})$. Then the second smallest eigenvector $\lambda$ of the normalized hypergraph Laplacian $\Pi^{-\frac{1}{2}} L \Pi^{-\frac{1}{2}}$ satisfies*

$$
\frac{\Phi(\mathcal{H})^2}{2} \leq \lambda \leq 2\Phi(\mathcal{H})
\tag{3}
$$

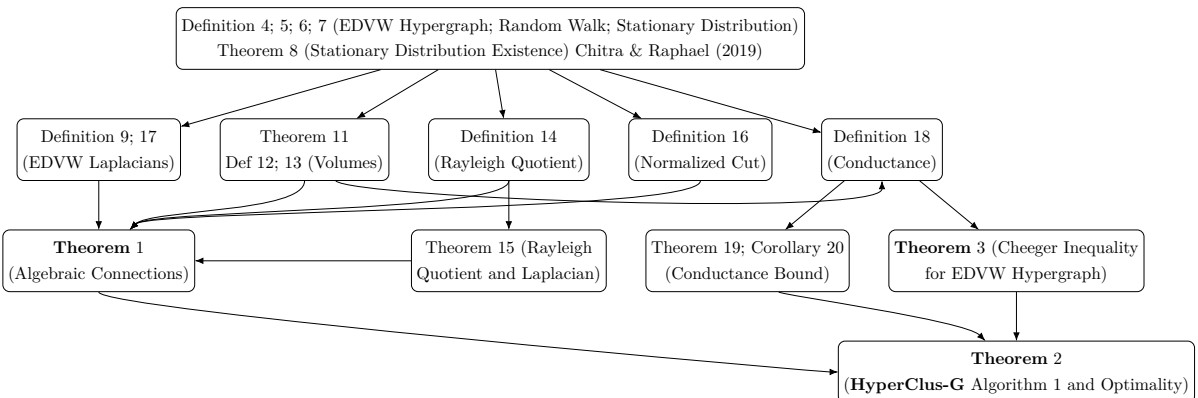

Figure 2: Logical flow of our work. "A->B" means A is required for developing B.

In fact, this theorem shows that **our HyperClus-G is also approximately linearly optimal in terms of conductance**. In other words, the conductance of the returned cluster $\Phi$ and the optimal conductance $\Phi^*$ satisfy $\Phi \leq O(\Phi^*)$. As shown in Table 2, we bridge the gap between the previous work Chitra & Raphael (2019) to broader spectral applications by further developing spectral theories on EDVW formulation.

**Technical Overview.** [reduced verbosity] The technical map is summarized in Figure 2. The key insight from the previous work (Chitra & Raphael, 2019) is to model the hypergraphs similar to directed graphs through the equivalence of random walks. Unlike classical graph theory, such directed graphs are edge-weighted, node-weighted, and contain self-loops. In this work, inspired by the definitions of Rayleigh Quotient, NCut, boundary/cut, volume, and conductance in graphs, we develop these definitions in the context of EDVW hypergraphs. We show that Theorem 1 and Theorem 3, properties that hold for graphs, still hold for hypergraphs using our unified definitions. From Theorem 3, we can further prove that our proposed HyperClus-G is approximately linearly optimal in terms of both NCut and conductance.

Our Appendix contains supplementary contents, such as trivial proofs and experimental details.

**Paper Organization.** [reduced verbosity] In Section 2, we introduce notations regarding EDVW hypergraphs. In Section 3, we introduce our definition of hypergraph Rayleigh Quotient and show its connection with the Laplacian and NCut. Then, we propose our HyperClus-G inspired by such a connection. In section 4, we prove the Cheeger Inequality of EDVW hypergraph, then show the optimality of our HyperClus-G. Finally, Section 5 presents comprehensive experiments to validate our theoretical findings.

## 2 Preliminaries

We use calligraphic letters (e.g., $\mathcal{A}$) for sets, capital letters for matrices (e.g., $A$), and unparenthesized superscripts to denote the power (e.g., $A^k$). For matrix indices, we use $A_{i,j}$ or $A(i,j)$ interchangeably to denote the entry in the $i^{th}$ row and the $j^{th}$ column. For row vector or column vector $v$, we use $v(i)$ to index its $i^{th}$ entry. Also, we denote hypergraph as $\mathcal{H}$ and graph as $\mathcal{G}$.

Table 3 contains important notation and hyperparameters for quick reference. A hypergraph consists of vertices and hyperedges. A hyperedge $e$ is a connection between two or more vertices. We use the notation $v \in e$ if the hyperedge $e$ connects vertex $v$. This is also called "$e$ is incident to $v$". Definition 4 and 5 provide necessary notations to define the hypergraph random walk in Definition 6. The transition matrix $P$ of EDVW hypergraphs is consistent with that of graphs.

**Definition 4.** *(Chitra & Raphael, 2019) (EDVW hypergraph). A hypergraph $\mathcal{H} = (\mathcal{V}, \mathcal{E}, \omega, \gamma)$ with edge-dependent vertex weight is defined as a set of vertices $\mathcal{V}$, a set $\mathcal{E} \subseteq 2^{\mathcal{V}}$ of hyperedges, a weight mapping $\omega(e) : \mathcal{E} \to \mathbb{R}_+$ on every hyperedge $e \in \mathcal{E}$, and weight mappings $\gamma_e(v) : \mathcal{V} \to \mathbb{R}_{\geq 0}$ corresponding to e on every*

Table 3: Table of Notation

| Symbol | Definition and Description |
|---|---|
| $\mathcal{H} = (\mathcal{V}, \mathcal{E}, \omega, \gamma)$ | hypergraph being investigated, with vertex set $\mathcal{V}$, hyperedge set $\mathcal{E}$, edge weight mapping $\omega$ and edge-dependent vertex weight mapping $\gamma$ |
| $n = |\mathcal{V}|$ | number of vertices in Hypergraph $\mathcal{H}$ |
| $m$ | number of hyperedge-vertex connections in Hypergraph $\mathcal{H}$, $m = \sum_{e \in \mathcal{E}} |e|$ |
| $d(v)$ | degree of vertex $v$, $d(v) = \sum_{e \in E(v)} w(e)$ |
| $\delta(e)$ | degree of hyperedge $e$, $\delta(e) = \sum_{v \in e} \gamma_e(v)$ |
| $R$ | $|\mathcal{E}| \times |\mathcal{V}|$ vertex-weight matrix |
| $W$ | $|\mathcal{V}| \times |\mathcal{E}|$ hyperedge-weight matrix |
| $D_{\mathcal{V}}$ | $|\mathcal{V}| \times |\mathcal{V}|$ vertex-degree matrix |
| $D_{\mathcal{E}}$ | $|\mathcal{E}| \times |\mathcal{E}|$ hyperedge-degree matrix |
| $P$ | $|\mathcal{V}| \times |\mathcal{V}|$ transition matrix of random walk on $\mathcal{H}$ |
| $\phi$ | $1 \times |\mathcal{V}|$ stationary distribution of random walk |
| $\Pi$ | $|\mathcal{V}| \times |\mathcal{V}|$ diagonal stationary distribution matrix |
| $L$ | $|\mathcal{V}| \times |\mathcal{V}|$ random-walk-based Laplacian |
| $p$ | $1 \times |\mathcal{V}|$ probability distribution on $\mathcal{V}$ |

vertex $v$. For $e_1 \neq e_2$, $\gamma_{e_1}(v)$ and $\gamma_{e_2}(v)$ may be different. Without loss of generality, we index the vertices by $1, 2, ..., |\mathcal{V}|$, and let $\mathcal{V} = \{1, 2, ..., |\mathcal{V}|\}$.

For an EDVW hypergraph $\mathcal{H} = (\mathcal{V}, \mathcal{E}, \omega, \gamma)$, $\omega(e) > 0$ for any $e \in \mathcal{E}$. $\gamma_e(v) \geq 0$ for any $e \in \mathcal{E}$ and $v \in \mathcal{V}$. Moreover, $\gamma_e(v) > 0 \iff v \in e$. For instance, in a citation hypergraph, each publication is captured by a hyperedge. While each publication may have different citations (i.e., edge-weight $w(e)$), each author may have individual weight of contributions (i.e., publication-dependent $\gamma_e(v)$).

**Definition 5.** *(Chitra & Raphael, 2019) (Vertex-weight matrix, hyperedge-weight matrix, vertex-degree matrix, and hyperedge-degree matrix of an EDVW hypergraph). $E(v) = \{e \in \mathcal{E} \text{ s.t. } v \in e\}$ is the set of hyperedges incident to vertex $v$. $d(v) = \sum_{e \in E(v)} w(e)$ denotes the degree of vertex $v$. $\delta(e) = \sum_{v \in e} \gamma_e(v)$ denotes the degree of hyperedge $e$. The vertex-weight matrix $R$ is an $|\mathcal{E}| \times |\mathcal{V}|$ matrix with entries $R(e, v) = \gamma_e(v)$. The hyperedge-weight matrix $W$ is a $|\mathcal{V}| \times |\mathcal{E}|$ matrix with entries $W(v, e) = \omega(e)$ if $v \in e$, and $W(v, e) = 0$ otherwise. The vertex-degree matrix $D_{\mathcal{V}}$ is a $|\mathcal{V}| \times |\mathcal{V}|$ diagonal matrix with entries $D_{\mathcal{V}} = d(v)$. The hyperedge-degree matrix $D_{\mathcal{E}}$ is a $|\mathcal{E}| \times |\mathcal{E}|$ diagonal matrix with entries $D_{\mathcal{E}}(e, e) = \delta(e)$.*

**Assumption 1.** *Since we are dealing with clustering, without loss of generality, we assume the hypergraph $\mathcal{H}$ is connected. A rigorous definition of hypergraph connectivity is provided in Appendix A.1.*

The *random walk* on EDVW hypergraph was proposed in (Chitra & Raphael, 2019). Intuitively, at time $t$, a random walker at vertex $u$ will first pick an edge $e$ incident to $u$ with the probability $\frac{\omega(e)}{d(u)}$, then pick a vertex $v$ from the picked edge $e$ with the probability $\frac{\gamma_e(v)}{\delta(e)}$, and finally move to vertex $v$ at time $t+1$. We can define the *transition matrix* $P$ to be a $|\mathcal{V}| \times |\mathcal{V}|$ matrix with entries $P_{u,v}$ to be the transition probability from $u$ to $v$ as follows.

**Definition 6.** *(Chitra & Raphael, 2019) (Hypergraph random walk). A random walk on a hypergraph with edge-dependent vertex weights $\mathcal{H} = (\mathcal{V}, \mathcal{E}, \omega, \gamma)$ is a Markov Chain on $\mathcal{V}$ with transition probabilities*

$$P_{u,v} = \sum_{e \in E(u)} \frac{\omega(e)}{d(u)} \frac{\gamma_e(v)}{\delta(e)} \tag{4}$$

*$P$ can be written in matrix form as $P = D_{\mathcal{V}}^{-1} W D_{\mathcal{E}}^{-1} R$ and it has row sum of 1. (Proof in Appendix A.2)*

Random walks and stationary distribution serve as a basis when conducting spectral analysis. Under assumption 1, a unique stationary distribution exists (referring to Theorem 8).

**Definition 7.** *(Stationary distribution of hypergraph random walk). The stationary distribution of the random walk with transition matrix $P$ is a $1 \times |\mathcal{V}|$ row vector $\phi$ such that*

$$\phi P = \phi; \ \phi(u) > 0 \ \forall u \in \mathcal{V}; \ \sum_{u \in \mathcal{V}} \phi(u) = 1 \tag{5}$$

*From $\phi$, define the stationary distribution matrix to be a $|\mathcal{V}| \times |\mathcal{V}|$ diagonal matrix with entries $\Pi_{i,i} = \phi(i)$.*

**Theorem 8.** *(Chitra & Raphael, 2019). The stationary distribution $\phi$ of hypergraph random walk exists.*

Theorem 8 has been proved in (Chitra & Raphael, 2019). In this paper, we give a simplified proof in Appendix A.4.

**Definition 9.** *(Chitra & Raphael, 2019) (random-walk-based hypergraph Laplacian). Let $\mathcal{H} = (\mathcal{V}, \mathcal{E}, \omega, \gamma)$ be a hypergraph with edge-dependent vertex weight. Let $P$ be the transition matrix and $\Pi$ be the corresponding stationary distribution matrix. Then, the random-walk-based hypergraph Laplacian $L$ is*

$$L = \Pi - \frac{\Pi P + P^T \Pi}{2} \tag{6}$$

In this work, in Appendix A.5, we further show that $L$ is consistent with graph Laplacian in terms of eigensystem as evidence of the rationality of Definition 9.

## 3 Spectral Properties Inspire Global Partitioning

In this section, we first extend the spectral theory for EDVW hypergraphs, then we introduce our HyperClus-G algorithm. Specifically, we show that using our definitions, there are algebraic connections among hypergraph NCut, Rayleigh Quotient, and Laplacian (Theorem 1), which are consistent with pair-wise graphs.

From Definition 10 to Definition 13, in the context of EDVW hypergraph, we re-define the volume of boundaries and vertex sets. We show that our definitions have properties that are consistent with those on graphs. Given a vertex set $\mathcal{S} \subseteq \mathcal{V}$, we use $\bar{\mathcal{S}}$ to denote its *complementary set*, where $\mathcal{S} \cup \bar{\mathcal{S}} = \mathcal{V}$ and $\mathcal{S} \cap \bar{\mathcal{S}} = \emptyset$. We have the following definition regarding the probability of a set.

**Definition 10.** *(Probability of a set). For a distribution $p$ on the vertices such that $\forall v \in \mathcal{V}, p(v) \geq 0$ and $\sum_{v \in \mathcal{V}} p(v) = 1$, we denote*

$$p(S) = \sum_{x \in S} p(x), \forall S \subseteq V \tag{7}$$

Equivalently, we can regard $p$ as a $1 \times |\mathcal{V}|$ vector with $p_i = p(i)$. From this definition, $\phi(\mathcal{S}) + \phi(\bar{\mathcal{S}}) = 1$. By definition of random walk and its stationary distribution,

$$\phi(S) = (\phi P)(S) = \phi(S) - \sum_{u \in \mathcal{S}, v \in \bar{\mathcal{S}}} \phi(u) P_{u,v} + \sum_{u \in \bar{\mathcal{S}}, v \in \mathcal{S}} \phi(u) P_{u,v} \tag{8}$$

Therefore, we have the following Theorem 11 that, in the stationary state, for any set $\mathcal{S}$, the probability of walking into $\mathcal{S}$ or out of $\mathcal{S}$ are the same.

**Theorem 11.** *Let $\mathcal{H} = (\mathcal{V}, \mathcal{E}, \omega, \gamma)$ be a hypergraph with edge-dependent vertex weights. Let $P$ be the transition matrix and $\phi$ be the corresponding stationary distribution. Then, for any vertex set $\mathcal{S} \subseteq \mathcal{V}$,*

$$\sum_{u \in \mathcal{S}, v \in \bar{\mathcal{S}}} \phi(u) P_{u,v} = \sum_{u \in \bar{\mathcal{S}}, v \in \mathcal{S}} \phi(u) P_{u,v} \tag{9}$$

For unweighted and undirected graphs, the volume of the *boundary/cut* of a partition is defined as $|\partial \mathcal{S}| = |\{\{x, y\} \in E | x \in \mathcal{S}, y \in \bar{\mathcal{S}}\}|$. The intuition behind is the symmetric property $|\partial \mathcal{S}| = |\partial \bar{\mathcal{S}}|$. Theorem 11 also describes such a property, and we find it intuitively suitable to be extended to the following definition.

**Definition 12.** *(Volume of hypergraph boundary). We define the volume of the hypergraph boundary, i.e., cut between $\mathcal{S}$ and $\bar{\mathcal{S}}$, by*

$$|\partial S| = \sum_{u \in \mathcal{S}, v \in \bar{\mathcal{S}}} \phi(u) P_{u,v} = \sum_{u \in \bar{\mathcal{S}}, v \in \mathcal{S}} \phi(u) P_{u,v} \tag{10}$$

*Furthermore, $0 \le |\partial \mathcal{S}| \le \sum_{u \in \mathcal{S}} \phi(u) \le 1$. $|\partial \mathcal{S}| = |\partial \bar{\mathcal{S}}|$.*

For unweighted, undirected graphs, the *volume of a vertex set $\mathcal{S}$* is defined as the degree sum of the vertices in $\mathcal{S}$. With the observation that $\phi(u) = \sum_{u \in \mathcal{S}, v \in \mathcal{V}} \phi(u) P_{u,v}$, $\phi(u)$ itself is already a sum of the transition probabilities and can be an analogy to vertex degree. We extend this observation to the following definition.

**Definition 13.** *(Volume of hypergraph vertex set). We define the volume of a vertex set $\mathcal{S} \subseteq \mathcal{V}$ in hypergraph $\mathcal{H}$ by*

$$vol(S) = \sum_{u \in \mathcal{S}} \phi(u) \in [0, 1] \tag{11}$$

Furthermore, we have $vol(\emptyset) = 0$, $vol(\mathcal{V}) = 1$, and $vol(\mathcal{S}) + vol(\bar{\mathcal{S}}) = 1$. Definition 12 and Definition 13 will serve as the basis of our unified formulation. By these two definitions, we also have $|\partial \mathcal{S}| \le vol(\mathcal{S})$, which is consistent with those on unweighted and undirected graphs.

Typically, for a graph $\mathcal{G}$ and its Laplacian $L_{\mathcal{G}} = D_{\mathcal{G}} - A_{\mathcal{G}}$, the unweighted Rayleigh Quotient is defined as a function $R_L : \mathbb{R}^n \setminus \{\mathbf{0}\} \to \mathbb{R}$ such that

$$R_{L_{\mathcal{G}}}(x) = \frac{x^T L_{\mathcal{G}} x}{x^T x} = \frac{\sum_{(i,j) \in E_{\mathcal{G}}} |x(i) - x(j)|^2}{\sum_{i \in V_{\mathcal{G}}} |x(i)|^2} \tag{12}$$

According to the form of generalized Rayleigh Quotient $R_L(x) = \frac{x^T L x}{x^T D x}$ (Golub & Loan, 1996), we extend this generalized Rayleigh Quotient to EDVW hypergraphs as follows.

**Definition 14.** *(Hypergraph Rayleigh Quotient). We define Rayleigh Quotient on $\mathcal{H}$ of any $|\mathcal{V}|$-dimensional real vector $x$ to be*

$$R(x) = \frac{\sum_{u,v} |x(u) - x(v)|^2 \phi(u) P_{u,v}}{\sum_u |x(u)|^2 \phi(u)} \tag{13}$$

We prove the following theorem which validates that our definition is consistent with the Rayleigh Quotient on graphs and satisfies the property similar to Equation 12.

**Theorem 15.** *For any $|\mathcal{V}|$-dimensional real vector $x$,*

$$R(x) = 2 \cdot \frac{x^T L x}{x^T \Pi x} = 2 \cdot \frac{< x^T L, x >}{x^T \Pi, x} \tag{14}$$

*(Proof in Appendix A.6)*

The 2-way Normalized Cut, a well-known measurement of the cluster quality, is defined as $\frac{|\partial \mathcal{S}|}{vol(\mathcal{S})} + \frac{|\partial \bar{\mathcal{S}}|}{vol(\bar{\mathcal{S}})}$. Analogically, we derive the NCut for EDVW hypergraphs using Definition 12.

**Definition 16.** *(Hypergraph Normalized Cut). For any vertex set $\mathcal{S} \subseteq \mathcal{V}$,*

$$NCut(\mathcal{S}, \bar{\mathcal{S}}) = \left( \frac{1}{vol(\mathcal{S})} + \frac{1}{vol(\bar{\mathcal{S}})} \right) \sum_{u \in \mathcal{S}, v \in \bar{\mathcal{S}}} \phi(u) P_{u,v} \tag{15}$$

Following this definition, the problem of global partitioning on EDVW hypergraphs aims to *find a vertex set $\mathcal{S}$ that minimizes the NCut value*. We now derive the association between our hypergraph Ncut, Rayleigh Quotient, and Laplacian, which directly inspires a spectral 2-way clustering approach (Algorithm 1).

**Theorem 1.** *(Restatement of Theorem 1 in Main Results) For any vertex set $\mathcal{S} \subseteq \mathcal{V}$, we define a $|\mathcal{V}|$-dimensional vector $x$ such that*

$$x(u) = \sqrt{\frac{vol(\bar{\mathcal{S}})}{vol(\mathcal{S})}}, \forall\, u \in \mathcal{S},$$

$$x(\bar{u}) = -\sqrt{\frac{vol(\mathcal{S})}{vol(\bar{\mathcal{S}})}}, \forall\, \bar{u} \in \bar{\mathcal{S}}. \tag{16}$$

*then* $NCut(\mathcal{S}, \bar{\mathcal{S}}) = \frac{1}{2} R(x) = \frac{x^T L x}{x^T \Pi x} \tag{17}$

*(Proof in Appendix A.7)*

By Theorem 1, the original global partitioning problem becomes

$$\min_x R(x) \text{ s.t. } x \text{ is defined as Eq 16 (fixed labeling index)} \tag{18}$$

The above is an NP-complete problem (Shi & Malik, 2000). If we relax the restriction of $x$ that each entry has to be either $\sqrt{\frac{vol(\bar{\mathcal{S}})}{vol(\mathcal{S})}}$ or $-\sqrt{\frac{vol(\mathcal{S})}{vol(\mathcal{S})}}$ for some $\mathcal{S}$, then the problem becomes

$$\min_x R(x) \Leftrightarrow \min_x \frac{x^T L x}{x^T \Pi x}, \text{ for } x \in \mathbb{R}^n \setminus \{\mathbf{0}\}. \tag{19}$$

Using the property of Rayleigh Quotient, the minimum can be achieved by choosing $x$ to be the eigenvector associated with the second smallest eigenvalue of the generalized eigenvalue system $Lx = \lambda \Pi x$ Ghojogh et al. (2019). Replacing $z = \Pi^{\frac{1}{2}} x$, we have

$$Lx = \lambda \Pi x \Leftrightarrow L\Pi^{-\frac{1}{2}} z = \lambda \Pi \Pi^{-\frac{1}{2}} z$$

$$\Leftrightarrow \Pi^{-\frac{1}{2}} L \Pi^{-\frac{1}{2}} z = \lambda z \tag{20}$$

$\Pi^{-\frac{1}{2}} L \Pi^{-\frac{1}{2}}$ is a symmetric and positive semi-definite matrix, for which all eigenvalues are non-negative real numbers. As the smallest eigenvalue is zero, and the associated clusters are trivial ($\mathcal{V}$ and $\emptyset$), we select the second smallest eigenvalue as the corresponding solution. Since the second smallest eigenvalue of $\Pi^{-\frac{1}{2}} L \Pi^{-\frac{1}{2}}$, denoted by $\lambda_1$, is real, there exists a real vector $z$ such that

$$\Pi^{-\frac{1}{2}} L \Pi^{-\frac{1}{2}} z = \lambda_1 z \tag{21}$$

We can then approximate the solution of the original global partitioning by choosing

$$\begin{cases} u \in S & \text{if } z(u) \geq 0 \\ u \in \bar{S} & \text{if } z(u) < 0 \end{cases} \tag{22}$$

Moreover, we define a new variant of hypergraph Laplacian.

**Definition 17.** *(Symmetric normalized random-walk-based hypergraph Laplacian). Let $\mathcal{H} = (\mathcal{V}, \mathcal{E}, \omega, \gamma)$ be a hypergraph with edge-dependent vertex weight. Let $P$ be the transition matrix and $\Pi$ be the corresponding stationary distribution matrix. Then, the symmetric normalized random-walk-based hypergraph Laplacian $L_{sym}$ is*

$$L_{sym} = \Pi^{-\frac{1}{2}} L \Pi^{-\frac{1}{2}} = I - \frac{\Pi^{\frac{1}{2}} P \Pi^{-\frac{1}{2}} + \Pi^{-\frac{1}{2}} P^T \Pi^{\frac{1}{2}}}{2} \tag{23}$$

$L_{sym}$ is also real and symmetric, and hence hermitian. The eigenvectors of $L_{sym}$ provide an approximate solution of 2-way hypergraph clustering. The formulation of this Laplacian is consistent with that for digraphs (Chung, 2005).

---

**Algorithm 1** HyperClus-G

---

**Require:** EDVW hypergraph $\mathcal{H} = (\mathcal{V}, \mathcal{E}, \omega, \gamma)$
**Ensure:** two clusters of vertices
  1: Compute $R, W, D_{\mathcal{V}}, D_{\mathcal{E}}$ according to Definition 5.
  2: Compute the transition matrix $P$ by Definition 6.
  3: Compute the stationary distribution by power iteration.
  4: Construct the stationary distribution matrix $\Pi$ and compute the Laplacian $L$ by Definition 9.
  5: Compute eigenvector of $\Pi^{-\frac{1}{2}} L \Pi^{-\frac{1}{2}}$ associated with the second smallest eigenvalue.
  6: Return the clusters based on the signs of the entries in the computed eigenvector.

---

## 4 Hypergraph Cheeger Inequality and Optimality of HyperClus-G

In this section, we show the hypergraph Cheeger Inequality (Theorem 3), which is consistent with the Cheeger Inequality of pair-wise graphs. The Cheeger Inequality requires the below definition of conductance. From the Cheeger Inequality, we prove the approximate linear optimality of HyperClus-G in terms of both NCut and Conductance.

**Definition 18.** *(Hypergraph conductance). The conductance of a cluster $\mathcal{S}$ on $\mathcal{H}$ is,*

$$
\begin{aligned}
\Phi(\mathcal{S}) &= \frac{|\partial \mathcal{S}|}{\min(vol(\mathcal{S}), vol(\bar{\mathcal{S}}))} \\
&= \frac{|\partial \mathcal{S}|}{\min(vol(\mathcal{S}), 1 - vol(\mathcal{S}))} \\
&= \frac{\sum_{u \in \mathcal{S}, v \in \bar{\mathcal{S}}} \phi(u) P_{u,v}}{\min(\sum_{v \in \mathcal{S}} \phi(u), 1 - \sum_{u \in \mathcal{S}} \phi(u))}
\end{aligned}
\tag{24}
$$

**Theorem 19.** *For any vertex set $\mathcal{S} \subseteq \mathcal{V}$, our hypergraph conductance $\Phi(\mathcal{S}) \in [0,1]$, which is consistent with graph conductance (Proof in Appendix A.8)*

### 4.1 Hypergraph Cheeger Inequality

Cheeger Inequality can be proved for our newly defined Symmetric normalized random-walk-based hypergraph Laplacian (Definition 17). The proof is similar to that for digraphs (Chung, 2005) and shows the rationality of our definitions.

**Theorem 3.** *(Hypergraph Cheeger Inequality, Restatement of Theorem 3 in Main Results) Let $\mathcal{H} = (\mathcal{V}, \mathcal{E}, \omega, \gamma)$ be an EDVW hypergraph. Define $\Phi(\mathcal{H}) = \min_{\mathcal{S} \subseteq \mathcal{V}} \Phi(\mathcal{S})$, where $\Phi(\mathcal{S})$ is defined in Definition 18. Then the second smallest eigenvector $\lambda$ of the normalized hypergraph Laplacian $L_{sym}$ satisfies*

$$
\frac{\Phi(\mathcal{H})^2}{2} \leq \lambda \leq 2\Phi(\mathcal{H})
\tag{25}
$$

*(Proof in Appendix A.9)*

### 4.2 Approximate Linear Optimality of HyperClus-G

**Optimality in terms of NCut.** From Theorem 1, and the relaxation of the restriction, the second smallest eigenvector $\lambda$ of the normalized hypergraph Laplacian $L_{sym}$ is an approximation of both optimal NCut and the NCut of the returned cluster. Therefore, the returned cluster is approximately linearly optimal in terms of NCut.

**Optimality in terms of conductance.** Using the RHS of Hypergraph Cheeger Inequality (Theorem 3), $\lambda$ satisfies $\lambda \leq 2\Phi(\mathcal{H})$, where $\Phi(\mathcal{H}) = \min_{\mathcal{S} \subseteq \mathcal{V}} \Phi(\mathcal{S})$ is the optimal conductance. Additionally, we have the following corollary from Definition 16 and 18, connecting NCut and conductance.

**Corollary 20.** *For any cluster $\mathcal{S}$, $\Phi(\mathcal{S}) \leq NCut(\mathcal{S}, \bar{\mathcal{S}})$.*

*Proof.* From Definition 16 and 18, [reduced space cost]

$$\Phi(\mathcal{S}) = \frac{|\partial \mathcal{S}|}{\min(vol(\mathcal{S}), vol(\bar{\mathcal{S}}))} = \frac{\sum_{u \in \mathcal{S}, v \in \bar{\mathcal{S}}} \phi(u) P_{u,v}}{\min(vol(\mathcal{S}), vol(\bar{\mathcal{S}}))}$$

$$\leq (\frac{1}{vol(\mathcal{S})} + \frac{1}{vol(\bar{\mathcal{S}})}) \sum_{u \in \mathcal{S}, v \in \bar{\mathcal{S}}} \phi(u) P_{u,v} = NCut(\mathcal{S}, \bar{\mathcal{S}}) \tag{26}$$

$\square$

Therefore, let the returned cluster to be $\mathcal{S}_{return}$, then we have

$$\Phi(\mathcal{S}_{return}) \leq NCut(\mathcal{S}_{return}, \bar{\mathcal{S}}_{return}) \approx \lambda \leq 2\Phi(\mathcal{H}) \tag{27}$$

which shows the approximately linear optimality of the returned partition from HyperClus-G. In fact, we can make the "$\approx$" more rigorous by observing that $\lambda = \min_{x \in \mathbb{R}^n \setminus \{\mathbf{0}\}} \frac{1}{2} R(x) \leq \frac{1}{2} R(\tilde{x}) = NCut(\mathcal{S}_{return}, \bar{\mathcal{S}}_{return})$, where $\tilde{x}$ is defined according to Theorem 1 with $\mathcal{S} = \mathcal{S}_{return}$. However, it is impossible to bound $NCut(\mathcal{S}_{\text{return}}, \bar{\mathcal{S}}_{\text{return}})$ in terms of big-$O$ of $\lambda$, since there exist counterexamples for sign-threshold spectral graph partitioning[3] (and thus also for EDVW hypergraphs) where $\lambda$ can be arbitrarily small while $NCut$ remains arbitrarily large.

## 5 Supportive Experiments

In this section, we demonstrate the effectiveness of our HyperClus-G on real-world data. We first describe the experimental settings and then discuss the experiment results. Details are provided in Appendix C.

### 5.1 Global Partitioning Setup

**Datasets and Hypergraph Constructions.** We use 9 datasets, all from the UC Irvine Machine Learning Repository. The datasets cover biology, computer science, business, and other subject areas. The construction progress of hypergraphs is the same as the seminal works (Zhou et al., 2006; Li & Milenkovic, 2018; Hein et al., 2013): for each dataset, each instance is a vertex, and we use a group of hyperedges to capture each feature. For example, in the Mushroom dataset, for its "stalk-shape" feature, we connect all the instances that have an enlarging stalk shape by a hyperedge, and connect all the instances that have a tapering stalk shape by another hyperedge. We use one hyperedge for each categorical feature. For numerical features, we first quantize them into bins of equal size, then map them to hyperedges. More details of each dataset and construction can be found in Appendix C.2.

**EDVW Assignments.** Each dataset contains at least 2 classes. For any dataset, assume set $\mathcal{V}_k$ contains all vertices in class $k$; for any hyperedge $e$, assume $\mathcal{V}_e$ contains all vertices that $e$ connects, then we can assign the deterministic EDVW:

$$\gamma_e(v) = |\mathcal{V}_k \cap \mathcal{V}_e|, \forall v \in \mathcal{V}_k \tag{28}$$

In other words, for any hyperedge $e$ and any vertex $v$ in class $k$, $\gamma_e(v)$ is the number of vertices in class $k$ that $e$ connects. This assignment makes the vertex weights for different hyperedges highly different. We intentionally and implicitly encode the label information into the vertex weights to validate that our HyeprClus-G can take the fine-grained information that is usually ignored by other hypergraph modeling methods.

**Settings and Metrics.** Among the 9 datasets, 5 of them have 2 classes, and 4 of them have at least 3 classes. For 2-class datasets, we apply HyperClus-G to partition the whole EDVW hypergraph into 2 clusters. For k-class datasets ($k \geq 3$), we call HyperClus-G iteratively for $k - 1$ times to get a $k$-way clustering. We first measure the quality of clusters by NCut. The 2-way clustering NCut has been provided in Definition 16, and we further define the $k$-way clustering NCut here.

---

[3]For example, dumbbell graphs consisting of two large cliques (or expanders) connected by a single edge.

Table 4: NCut Comparison($\downarrow$) of Global Partitioning Task on EDVW Hypergraphs.

| Method Name | 2-way Clustering | | | | | $k$-way Clustering (k $\geq$ 3) | | | |
| --- | --- | --- | --- | --- | --- | --- | --- | --- | --- |
| | Mushroom | Rice | Car | Digit-24 | Covertype | Zoo | Wine | Letter | Digit |
| STAR++ | 0.6484 | **0.3479** | 0.8347 | 0.6458 | 0.8912 | 5.1402 | 1.5879 | 2.1866 | 8.4951 |
| CLIQUE++ | 0.6426 | 0.4041 | 0.8347 | 0.6445 | **0.8887** | 5.1478 | 1.6742 | 2.2177 | 8.4384 |
| DiffEq | 0.9425 | 0.7445 | 0.9729 | 0.9611 | 0.9895 | 5.8772 | 1.9667 | 2.8406 | 8.9477 |
| node2vec | 0.8560 | 0.7596 | 0.9334 | 0.6417 | 0.9676 | 5.5557 | 1.8283 | 2.1663 | 8.4745 |
| hyperedge2vec | 0.6656 | 0.3936 | 0.8360 | 0.6456 | 0.9102 | 5.1443 | 1.6203 | 2.1258 | 8.4517 |
| actual class label | 0.6778 | 0.3801 | 0.8490 | 0.6415 | 0.9300 | 5.4676 | 1.9883 | 2.5765 | 8.7046 |
| HyperClus-G (Ours) | **0.6388** | 0.3577 | **0.8320** | **0.6412** | 0.8911 | **5.1386** | **1.5831** | **2.1184** | **8.4197** |

Table 5: F1s Comparison($\uparrow$) of Global Partitioning Task on EDVW Hypergraphs.

| Method Name | 2-way Clustering | | | | | $k$-way Clustering (k $\geq$ 3) | | | |
| --- | --- | --- | --- | --- | --- | --- | --- | --- | --- |
| | Mushroom | Rice | Car | Digit-24 | Covertype | Zoo | Wine | Letter | Digit |
| STAR++ | 0.903, 0.874 | 0.900, 0.855 | 0.569, 0.550 | 0.980, 0.980 | 0.694, 0.436 | 0.880 | 0.385 | 0.626 | 0.514 |
| CLIQUE++ | 0.898, 0.870 | 0.854, 0.843 | 0.569, 0.550 | 0.983, 0.983 | 0.701, 0.455 | 0.706 | 0.365 | 0.531 | **0.633** |
| DiffEq | 0.691, 0.247 | 0.765, 0.344 | 0.653, 0.154 | 0.687, 0.278 | 0.837, 0.158 | 0.313 | 0.369 | 0.275 | 0.148 |
| node2vec | 0.551, 0.508 | 0.663, 0.148 | 0.640, 0.541 | 0.995, 0.995 | 0.688, 0.064 | 0.452 | 0.300 | 0.647 | 0.545 |
| hyperedge2vec | 0.844, 0.835 | 0.843, 0.808 | 0.571, 0.545 | 0.971, 0.970 | 0.677, 0.280 | 0.861 | 0.393 | 0.587 | 0.451 |
| HyperClus-G (Ours) | **0.915, 0.889** | **0.947, 0.932** | **0.706, 0.697** | **0.996, 0.996** | **0.701, 0.488** | **0.893** | **0.395** | **0.704** | 0.629 |

**Definition 21.** *For any k-way clustering of vertex set $\mathcal{V}$,*

$$NCut(\mathcal{S}_1, ..., \mathcal{S}_k) = \sum_{i=1}^{k} \frac{|\partial \mathcal{S}_i|}{vol(\mathcal{S}_i)} \tag{29}$$

Another metric on cluster quality is whether the partition aligns with the "ground-truth" label data. The vertices in the same cluster are determined by the algorithms to be closely related in structure, while the vertices in the same label class should also be internally similar. These two similarities usually align, and therefore we greedily match the clusters with classes and see if each matching has a high F1 score. For 2-way clustering, we report the F1 scores of both clusters. For $k$-way clustering, we report the weighted F1 scores of all clusters. More details are provided in Appendix C.3.

**Baselines.** We compare our HyperClus-G with multiple state-of-the-art methods. STAR++ and CLIQUE++ expansions convert the hypergraph into weighted graphs and then adopt spectral clustering on graphs. Moreover, since it is proven that, the EIVW Laplacian proposed by (Zhou et al., 2006) is equal to the Laplacian of the corresponding STAR++ expanded graph (Agarwal et al., 2006; Chitra & Raphael, 2019), our STAR++ baseline is equivalent to spectral clustering on EIVW hypergraphs by ignoring the assigned EDVW. DiffEq (Takai et al., 2020) is based on differential operators. node2vec (Grover & Leskovec, 2016) and event2vec (a.k.a., hyperedge2vec) (Fu et al., 2019) are two unsupervised graph embedding methods. We first convert the hypergraph into the required input graph form, then call the k-means algorithm after we get the vertex embedding. We also compare with the NCut of the actual labeled classes ($\mathcal{V}_1, ..., \mathcal{V}_k$ partitioning). For the nondeterministic baselines, we run the experiment 10 times and report the mean. The standard deviations are relatively small, and we put them in Appendix C.6.1. More details on baselines are provided in Appendix C.5.

## 5.2 Results

The global partitioning experiment results are shown in Tables 4 and 5. The best results are marked in bold, and the second-best results are marked with underlines. For 2-way clustering, we also consider that the two F1s of a high-quality partition should be fairly similar. **First**, our algorithm HyperClus-G generally outperforms the baseline methods in terms of NCut. It is worth noting that, facing a combinatorial optimization problem, our HyperClus-G constantly outperforms the actual-class partition, and may already get very close to the optimal NCut. **Second**, STAR++ and CLIQUE++ expansions, as graph approximations

Table 6: Execution Time(↓) on Global Partitioning Task in seconds. Full Table in Appendix C.6.2.

| Method | 2-way Clustering | | | $k$-way Clustering (k ≥ 3) | | |
|---|---|---|---|---|---|---|
| | Mushroom | Rice | Covertype | Wine | Letter | Digit |
| STAR++ | 72.82 | 50.80 | 318.55 | 32.18 | 37.54 | 59.05 |
| CLIQUE++ | 216.04 | 13.32 | 1099.03 | 77.75 | 50.25 | 297.37 |
| HyperClus-G | **17.73** | **5.04** | **87.72** | **11.48** | **35.13** | **26.49** |

for hypergraphs, are very strong baselines and can generate sub-optimal clustering. **Third**, in terms of F1 scores matched with actual classes, our HyperClus-G also achieves the best performance in general. The only exception is Digit $k$-way clustering, where HyperClus-G achieves the second best, but is very close to the best. This again shows that STAR++ and CLIQUE++ expansions are strong baselines. Unsupervised embedding methods can beat STAR++ and CLIQUE++ expansions on smaller datasets. **Fourth**, though HyperClus-G is not designed for $k$-way clustering, applying it multiple times returns a high-quality $k$-way clustering. **Fifth**, it turns out that the second smallest eigenvalue of $L_{sym}$ is a good approximation for NCut (Refer to Section 5.3 for details), which is consistent with our theoretical analysis. **Sixth**, we report the execution time in Table 6. in fact, for NCut in Rice, Covertype, and F1 in Digit, where STAR++ or CLIQUE++ expansion achieves the best performance, it needs to take 10× time compared to HyperClus-G.

Table 7: Comparison of 2-way NCut and its Eigenvalue Approximation.

| Value | 2-way Clustering | | | | |
|---|---|---|---|---|---|
| | Mushroom | Rice | Car | Digit-24 | Covertype |
| 2-way NCut value of HyperClus-G results | 0.6388 | 0.3577 | 0.8320 | 0.6412 | 0.8911 |
| 2nd smallest eigenvalue of $L_{sym}$ (Definition 17) | 0.6171 | 0.2686 | 0.7655 | 0.6220 | 0.8663 |
| relative error $\|eigenvalue - ncut\| / ncut$ | 3.397% | 24.91% | 7.993% | 2.994% | 2.783% |

## 5.3 Validation of the Eigenvalue Approximation

According to our theoretical study of Theorem 1, and the later transformation of the problem, the second smallest eigenvalue of the symmetric normalized random-walk-based hypergraph Laplacian $L_{sym}$ should be an approximation of the NCut value by relaxing the combinatorial optimization constraint. We validate this by showing the 2-way NCut value of the clustering result from our algorithm HyperClus-G, and compare it with the second smallest eigenvalue of $L_{sym}$. The data are shown in Table 7. It turns out that the two numbers are positively correlated, and the error tends to be small. The Rice dataset, which has much more hyperedges than other datasets, has the largest relative error.

## 5.4 Ablation Study

To show that our algorithm exploits the EDVW information, we conduct an ablation study of STAR++, CLIQUE++, and HyperClus-G on the EIVW hypergraphs where the EDVW is set to be a constant value of one. From the results, we observe that the clustering performance of HyperClus-G on EIVW hypergraphs is the same as STAR++. First, this validates that the performance boost given EDVW hypergraphs is because our HyperClus-G can exploit EDVW. Second, the conjecture that our HyperClus-G degenerates to STAR++, given an all-one-EDVW hypergraph, is naturally raised, even though the dimensions of their Laplacians are different. Experimentally, we observe that for all-one-EDVW hypergraphs, the second smallest eigenvector of $L_{sym}$ has the same direction as the sub-vector of all the original vertices of the second smallest eigenvector of the random-walk Laplacian of the STAR++ graph. This is an interesting phenomenon that is worth studying in the future.

Table 8: NCut Comparison(↓) of Global Partitioning Task on EIVW Hypergraphs.

| Method Name | 2-way Clustering | | | | |
|---|---|---|---|---|---|
| | Mushroom | Rice | Car | Digit-24 | Covertype |
| STAR++ | 0.6926 | 0.3754 | 0.8340 | 0.7047 | 0.8884 |
| CLIQUE++ | 0.6870 | 0.4318 | 0.8340 | 0.7045 | 0.8943 |
| actual class label | 0.7554 | 0.4833 | 0.8782 | 0.7080 | 0.9339 |
| HyperClus-G (Ours) | 0.6926 | 0.3754 | 0.8340 | 0.7047 | 0.8884 |

Table 9: F1 Comparison(↑) of Global Partitioning Task on EIVW Hypergraphs.

| Method Name | 2-way Clustering | | | | |
|---|---|---|---|---|---|
| | Mushroom | Rice | Car | Digit-24 | Covertype |
| STAR++ | 0.903, 0.874 | 0.900, 0.855 | 0.569, 0.550 | 0.980, 0.980 | 0.694, 0.436 |
| CLIQUE++ | 0.898, 0.870 | 0.854, 0.843 | 0.569, 0.550 | 0.983, 0.983 | 0.701, 0.455 |
| HyperClus-G (Ours) | 0.903, 0.874 | 0.900, 0.855 | 0.569, 0.550 | 0.980, 0.980 | 0.694, 0.436 |

## 6 Conclusion

This work advances a unified random walk-based formulation of hypergraphs based on EDVW modeling. We introduce key definitions, such as Rayleigh Quotient, NCut, and conductance, aligning them with graph theory to advance spectral analysis. By establishing the relationship between the normalized hypergraph Laplacian and the NCut value, we develop the HyperClus-G algorithm for spectral clustering on EDVW hypergraphs. Our theoretical analysis and extensive experimental validation demonstrate that HyperClus-G achieves approximately optimal partitioning and outperforms existing methods in practice.

## Acknowledgment

This work is supported by National Science Foundation under Award No. IIS-2117902 and AFOSR (FA9550-24-1-0002). The views and conclusions are those of the authors and should not be interpreted as representing the official policies of the funding agencies or the government.

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

# A   Supplementary Definitions, Proof of Lemmas and Theorems

## A.1   Definition of Hyperpath and Hypergraph Connectivity

**Definition 22.** *(Zhou et al., 2006) (Hyperpath and hypergraph connectivity). We say there is a hyperpath between vertices $v_1$ and $v_k$ when there is a sequence of distinct vertices and hyperedges $v_1, e_1, v_2, e_2, ..., e_{k-1}, v_k$ such that $v_i \in e_i$ and $v_{i+1} \in e_i$ for $1 \le i \le k-1$. A hypergraph $\mathcal{H}$ is connected if there exists a hyperpath for any pair of vertices.*

## A.2   Proof of Definition 6 that P has row sum 1

*Proof.* The sum of the $v^{th}$ row of $P$ is:

$$\sum_u P_{v,u} = \sum_u \sum_{e \in E(v)} \frac{w(e)}{d(v)} \cdot \frac{\gamma_e(u)}{\delta(e)} \tag{30}$$

Since $H$ is connected, there exists $e \in E(v)$ and therefore,

$$\sum_u P_{v,u} = \sum_{e \in E(v)} \frac{w(e)}{d(v)} \cdot \sum_u \frac{\gamma_e(u)}{\delta(e)} = \sum_{e \in E(v)} \frac{w(e)}{d(v)} = 1 \tag{31}$$

$\square$

## A.3   $P$ is Irreducible

**Lemma 23.** *The transition matrix $P$ of hypergraph random walk defined in definition 6 is irreducible.*

*Proof.* First, we create a new matrix $Q$ by replacing $P_{u,v}$ by 1 if $P_{u,v} > 0$, and prove that $Q$ is irreducible. Hence, $P$ holds the same feature.

By the assumption that the undirected hypergraph $H$ is connected, every node $u$ is reachable from any node $v \ne u$ through a sequence of hyperedges. $Q_{u,v} = 1$ if and only if $P_{u,v} > 0$.

By the definition of $P$,

$$P_{u,v} = \sum_{e \in E(v)} \frac{\omega(e)}{d(u)} \frac{\gamma_e(v)}{\delta(e)} > 0 \tag{32}$$

if there is a hyperedge connecting $v$ and $u$.

Therefore, the directed graph $G$ with $Q$ as the adjacency matrix can be constructed by replacing every hyperedge with a directed fully-connected clique.

For any node pair $v, u$, by the assumption of $H$, there exists a sequence of hyperedges from $v$ to $u$; hence, in $G$, there also exists a sequence of directed edges from $v$ to $u$. This means $G$ is strongly connected, and $Q$ is irreducible. According to Theorem 6.2.44 in Matrix Analysis (Horn & Johnson, 2012), $P$ is irreducible. $\square$

## A.4   Proof of Theorem 8

*Proof.* According to the Perron-Frobenius Theorem and Lemma 23, the irreducible matrix $P$ has a unique left eigenvector with all entries positive. We denote and normalize this row vector as $\phi$ and prove $\phi P = \phi \iff P^T \phi^T = \phi^T$. According to the definition of $\phi$,

$$P^T \phi^T = \rho(P) \phi^T \tag{33}$$

where $\rho(P)$ is the spectral radius, which is the maximum eigenvalue of $\rho(P)$.

$$
\begin{aligned}
\rho(P) = \rho(P) \sum_{i=1}^{|V|} \phi^T(i) &= \sum_{i=1}^{|V|} \rho(P) \phi^T(i) \\
&= \sum_{i=1}^{|V|} \sum_{j=1}^{|V|} P_{ij}^T \phi^T(j) = \sum_{j=1}^{|V|} \phi^T(j) \sum_{i=1}^{|V|} P_{ij}^T \\
&= \sum_{j=1}^{|V|} \phi^T(j) \cdot 1 = 1.
\end{aligned}
\tag{34}
$$

Therefore, $\phi$ satisfies $P^T \phi = \phi$, $\forall u\ \phi(u) > 0$ and $\sum_u \phi(u) = 1$.

$\square$

## A.5 Additional Lemma to Support Random-walk-based Hypergraph Laplacian (Definition 9)

**Lemma 24.** *For a graph $\mathcal{G}$ with pair-wise relations, let $D$ be its degree matrix and $A$ be its adjacency matrix. Let $P_{\mathcal{G}} = D^{-1}A$ be the transition matrix of graph random walk, and $\Pi_{\mathcal{G}}$ be the stationary distribution of $P_{\mathcal{G}}$. Then $L = \Pi_{\mathcal{G}} - \frac{\Pi_{\mathcal{G}} P_{\mathcal{G}} + P_{\mathcal{G}}^T \Pi_{\mathcal{G}}}{2}$ has the same eigenvectors as the graph Laplacian $L_{\mathcal{G}} = D - A$.*

*Proof.* This lemma serves as an evidence that

$$
L = \Pi - \frac{\Pi P + P^T \Pi}{2}
\tag{35}
$$

can be degenerated into unweighted non-hyper graphs (graphs with only pair-wise relations). For pair-wise graphs, given $P_{\mathcal{G}} = (AD^{-1})^T = D^{-1}A^T = D^{-1}A$, we have $\phi_{\mathcal{G}}(u) = \frac{d(u)}{\sum_v d(v)}$ because,

$$
(P_{\mathcal{G}}^T \phi_{\mathcal{G}})(i) = \sum_{j=1}^{|V|} P_{\mathcal{G}}(i,j)^T \phi_{\mathcal{G}}(j) = \sum_{j=1}^{|V|} \frac{A_{ij}}{d(j)} \cdot \frac{d(j)}{\sum_v d(v)} = \frac{\sum_{j=1}^{|V|} A_{ij}}{\sum_v d(v)} = \frac{d(i)}{\sum_v d(v)}
\tag{36}
$$

Hence,

$$
\Pi_{\mathcal{G}} = \frac{D}{\sum_v d(v)}.
\tag{37}
$$

Therefore, for regular graphs, if we follow the same definition of hypergraph Laplacian as of equation 35,

$$
L = \frac{D}{\sum_v d(v)} - \frac{DD^{-1}A + AD^{-1}D}{2\sum_v d(v)} = \frac{D - A}{\sum_v d(v)},
\tag{38}
$$

which has the same eigenvectors as the regular graph Laplacian $D - A$.

$\square$

## A.6 Proof of Theorem 15

*Proof.* Notice that when $x^T A x$ is a scalar,

$$
x^T A x = (x^T A x)^T = x^T A^T x
\tag{39}
$$

substituting $A = \Pi P$, we have

$$
x^T \Pi P x = x^T P^T \Pi x
\tag{40}
$$

Also note that

$$
\begin{aligned}
\sum_{u,v} x^2(u)\phi(u)P_{u,v} &= \sum_u x^2(u)\phi(u)\sum_v P_{u,v} = \sum_u x^2(u)\phi(u) \\
\sum_{u,v} x^2(v)\phi(u)P_{u,v} &= \sum_v x^2(v)\sum_u \phi(u)P_{u,v} \overset{Definition\ 7}{=} \sum_v x^2(v)\phi(v)
\end{aligned}
\tag{41}
$$

Then,

$$
\begin{aligned}
R(x) &= \frac{\sum_{u,v}(x(u)-x(v))^2\phi(u)P_{u,v}}{x^T\Pi x} \\
&= \frac{\sum_{u,v}x^2(u)\phi(u)P_{u,v}+\sum_{u,v}x^2(v)\phi(u)P_{u,v}-2\sum_{u,v}x(u)x(v)\phi(u)P_{u,v}}{x^T\Pi x} \\
&\overset{Equation\ 41}{=} \frac{\sum_u x^2(u)\phi(u)+\sum_v x^2(v)\phi(v)-2\cdot x^T\Pi Px}{x^T\Pi x} \\
&\overset{Equation\ 40}{=} \frac{2x^T\Pi x - 2\cdot\frac{1}{2}(x^T\Pi Px + x^T P^T\Pi x)}{x^T\Pi x} \\
&= 2\cdot\frac{x^T(\Pi - \frac{\Pi P+P^T\Pi}{2})x}{x^T\Pi x} \\
&\overset{Definition\ 9}{=} 2\cdot\frac{x^T L x}{x^T\Pi x}.
\end{aligned}
\tag{42}
$$

$\square$

## A.7  Proof of Theorem 1

*Proof.* For a partition $\mathcal{S}\cup\bar{\mathcal{S}}=\mathcal{V}, \mathcal{S}\cap\bar{\mathcal{S}}=\emptyset$, we have a $|V|\times 1$ vector $x(u)$, where

$$
x(u)=\begin{cases}
\sqrt{\frac{vol(\bar{\mathcal{S}})}{vol(\mathcal{S})}} & \text{if } u\in\mathcal{S} \\
-\sqrt{\frac{vol(\mathcal{S})}{vol(\bar{\mathcal{S}})}} & \text{if } u\in\bar{\mathcal{S}}
\end{cases}
\tag{43}
$$

From the definition of $NCut$ in Eq 16, we can compute $R(x)$ as follows:

$$
\begin{aligned}
R(x) &= \frac{\sum_{u,v} |x(u) - x(v)|^2 \, \phi(u) P_{u,v}}{\sum_u |x(u)|^2 \cdot \phi(u)} \\[8pt]
&= \frac{\sum_{(u \in \mathcal{S}, v \in \bar{\mathcal{S}}) or (u \in \bar{\mathcal{S}}, v \in \mathcal{S})} (\frac{vol(\bar{\mathcal{S}})}{vol(\mathcal{S})} + \frac{vol(\mathcal{S})}{vol(\bar{\mathcal{S}})} + 2) \cdot \phi(u) P_{u,v}}{\sum_{u \in \mathcal{S}} \frac{vol\bar{\mathcal{S}}}{vol\mathcal{S}} \cdot \phi(u) + \sum_{u \in \bar{\mathcal{S}}} \frac{vol\mathcal{S}}{vol\bar{\mathcal{S}}} \cdot \phi(u)} \\[8pt]
&= \frac{(\frac{vol\bar{\mathcal{S}} + vol\mathcal{S}}{vol\mathcal{S}} + \frac{vol\mathcal{S} + vol\bar{\mathcal{S}}}{vol\bar{\mathcal{S}}}) \sum_{(u \in \mathcal{S}, v \in \bar{\mathcal{S}}) or (u \in \bar{\mathcal{S}}, v \in \mathcal{S})} \cdot \phi(u) \cdot P_{u,v}}{\frac{vol\bar{\mathcal{S}}}{vol\mathcal{S}} \sum_{u \in \mathcal{S}} \phi(u) + \frac{vol\mathcal{S}}{vol\bar{\mathcal{S}}} \sum_{u \in \mathcal{S}} \phi(u)} \\[8pt]
&= \frac{(\frac{vol\bar{\mathcal{S}} + vol\mathcal{S}}{vol\mathcal{S}} + \frac{vol\mathcal{S} + vol\bar{\mathcal{S}}}{vol\bar{\mathcal{S}}})(\sum_{u \in \mathcal{S}, v \in \bar{\mathcal{S}}} \cdot \phi(u) \cdot P_{u,v} + \sum_{u \in \bar{\mathcal{S}}, v \in \mathcal{S}} \cdot \phi(u) \cdot P_{u,v})}{\frac{vol\bar{\mathcal{S}}}{vol\mathcal{S}} \sum_{u \in \mathcal{S}} \phi(u) + \frac{vol\mathcal{S}}{vol\bar{\mathcal{S}}} \sum_{u \in \mathcal{S}} \phi(u)} \\[8pt]
&= \frac{(\frac{vol\bar{\mathcal{S}} + vol\mathcal{S}}{vol\mathcal{S}} + \frac{vol\mathcal{S} + vol\bar{\mathcal{S}}}{vol\bar{\mathcal{S}}}) 2 \sum_{u \in \mathcal{S}, v \in \bar{\mathcal{S}}} \cdot \phi(u) \cdot P_{u,v}}{\frac{vol\bar{\mathcal{S}}}{vol\mathcal{S}} \sum_{u \in \mathcal{S}} \phi(u) + \frac{vol\mathcal{S}}{vol\bar{\mathcal{S}}} \sum_{u \in \mathcal{S}} \phi(u)} \\[8pt]
&= \frac{2(vol\bar{\mathcal{S}} + vol\mathcal{S})(\frac{1}{vol\mathcal{S}} + \frac{1}{vol\bar{\mathcal{S}}}) \sum_{u \in \mathcal{S}, v \in \bar{\mathcal{S}}} \cdot \phi(u) \cdot P_{u,v}}{\frac{vol\bar{\mathcal{S}}}{vol\mathcal{S}} \sum_{u \in \mathcal{S}} \phi(u) + \frac{vol\mathcal{S}}{vol\bar{\mathcal{S}}} \sum_{u \in \mathcal{S}} \phi(u)} \\[8pt]
&= \frac{2(vol\bar{\mathcal{S}} + vol\mathcal{S}) NCut(\mathcal{S}, \bar{\mathcal{S}})}{vol\bar{\mathcal{S}} + vol\mathcal{S}} \\[8pt]
&= 2 NCut(\mathcal{S}, \bar{\mathcal{S}})
\end{aligned}
\tag{44}
$$

$\square$

## A.8   Proof of Theorem 19

*Proof.* Recall from the definition 18, 12 and 13, the numerator and denominator are both non-negative, and

$$
\Phi(\mathcal{S}) = \frac{|\partial \mathcal{S}|}{\min(vol(\mathcal{S}), vol(\bar{\mathcal{S}}))} = \frac{|\partial \bar{\mathcal{S}}|}{\min(vol(\mathcal{S}), vol(\bar{\mathcal{S}}))}
\tag{45}
$$

Since $|\partial \mathcal{S}| \le vol(\mathcal{S}), |\partial \bar{\mathcal{S}}| \le vol(\bar{\mathcal{S}})$, $0 \le \Phi(\mathcal{S}) \le \frac{\min(vol(\mathcal{S}), vol(\bar{\mathcal{S}}))}{\min(vol(\mathcal{S}), vol(\bar{\mathcal{S}}))} = 1$.  $\square$

## A.9   Proof of Theorem 3

*Proof.* From Theorem 15,

$$
\lambda = \min_{x \in \mathbb{R}^n \setminus \{\mathbf{0}\}} \frac{x^T L x}{x^T \Pi x} = \min_{x \in \mathbb{R}^n \setminus \{\mathbf{0}\}} \frac{1}{2} R(x)
\tag{46}
$$

Let $\mathcal{S} = \arg\min_{\mathcal{S}' \subseteq \mathcal{V}} \Phi(\mathcal{S}')$, $y(u) = \begin{cases} \frac{1}{vol(\mathcal{S})}, & \text{if } u \in \mathcal{S} \\ -\frac{1}{1 - vol(\mathcal{S})}, & \text{otherwise} \end{cases}$ , then continue with Equation 46,

$$\lambda = \min_{x \in \mathbb{R}^n \setminus \{\mathbf{0}\}} \frac{1}{2} R(x)$$

$$\leq \frac{1}{2} R(y)$$

$$= \frac{1}{2} \frac{\sum_{u,v} |y(u) - y(v)|^2 \phi(u) P_{u,v}}{\sum_u |y(u)|^2 \phi(u)}$$

$$= \frac{1}{2} \frac{\sum_{u \in \mathcal{S}, v \in \bar{\mathcal{S}}} |y(u) - y(v)|^2 \phi(u) P_{u,v} + \sum_{u \in \bar{\mathcal{S}}, v \in \mathcal{S}} |y(u) - y(v)|^2 \phi(u) P_{u,v}}{\sum_{u \in \mathcal{S}} |y(u)|^2 \phi(u) + \sum_{u \in \bar{\mathcal{S}}} |y(u)|^2 \phi(u)}$$

$$= \frac{1}{2} \frac{\sum_{u \in \mathcal{S}, v \in \bar{\mathcal{S}}} (\frac{1}{vol(\mathcal{S})} + \frac{1}{1 - vol(S)})^2 \phi(u) P_{u,v} + \sum_{u \in \bar{\mathcal{S}}, v \in \mathcal{S}} (\frac{1}{1 - vol(S)} + \frac{1}{vol(S)})^2 \phi(u) P_{u,v}}{\sum_{u \in \mathcal{S}} |\frac{1}{vol(\mathcal{S})}|^2 \phi(u) + \sum_{u \in \bar{\mathcal{S}}} |\frac{1}{1 - vol(\mathcal{S})}|^2 \phi(u)} \tag{47}$$

$$= (\frac{1}{vol(\mathcal{S})} + \frac{1}{1 - vol(S)})^2 \frac{\frac{1}{2}(\sum_{u \in \mathcal{S}, v \in \bar{\mathcal{S}}} \phi(u) P_{u,v} + \sum_{u \in \bar{\mathcal{S}}, v \in \mathcal{S}} \phi(u) P_{u,v})}{(\frac{1}{vol(\mathcal{S})})^2 \sum_{u \in \mathcal{S}} \phi(u) + (\frac{1}{1 - vol(\mathcal{S})})^2 \sum_{u \in \bar{\mathcal{S}}} \phi(u)}$$

$$= (\frac{1}{vol(\mathcal{S})} + \frac{1}{1 - vol(S)})^2 \frac{|\partial \mathcal{S}|}{(\frac{1}{vol(\mathcal{S})})^2 vol(\mathcal{S}) + (\frac{1}{1 - vol(\mathcal{S})})^2 (1 - vol(\mathcal{S}))}$$

$$= \frac{|\partial \mathcal{S}|}{vol(\mathcal{S})(1 - vol(\mathcal{S}))}$$

$$\leq \frac{2|\partial \mathcal{S}|}{\min(vol(\mathcal{S}), 1 - vol(\mathcal{S}))}$$

$$= 2\Phi(\mathcal{H})$$

Thus, the LFS has been proven. For the RHS, assume $w$ is the eigenvector of the normalized Laplacian associated with $\lambda$, let vector $f = \Pi^{-\frac{1}{2}} w$. Then,

$$\lambda f(u) \phi(u) = \lambda (\Pi^{-\frac{1}{2}} w)(u) \phi(u) = (\Pi^{-\frac{1}{2}} L_{sym} w)(u) \phi(u) = (\Pi \Pi^{-\frac{1}{2}} L_{sym} w)(u)$$

$$= (\Pi^{\frac{1}{2}} L_{sym} \Pi^{\frac{1}{2}} f)(u) = (Lf)(u) = [(\Pi - \frac{\Pi P + P^T \Pi}{2}) f](u)$$

$$= \phi(u) f(u) - \frac{\sum_v f(v) \phi(u) P_{u,v}}{2} - \frac{\sum_v f(v) \phi(v) P_{v,u}}{2} \tag{48}$$

$$= \frac{1}{2} \sum_v (f(u) - f(v))(\phi(u) P_{u,v} + \phi(v) P_{v,u})$$

Therefore, $\forall u$

$$\lambda = \frac{\sum_v (f(u) - f(v))(\phi(v) P_{v,u} + \phi(u) P_{u,v})}{2 f(u) \phi(u)} \tag{49}$$

Let $\mathcal{V}_+ = \{u : f(u) \geq 0\}$ and let vector $g(u) = \begin{cases} f(u), \text{ if } f(u) \geq 0 \\ 0, \text{ otherwise} \end{cases}$, then

$$\lambda = \frac{\sum_{u \in \mathcal{V}_+} f(u) \sum_v (f(u) - f(v))(\phi(v) P_{v,u} + \phi(u) P_{u,v})}{\sum_{u \in \mathcal{V}_+} f(u) 2 f(u) \phi(u)}$$

$$= \frac{\sum_{u \in \mathcal{V}} g(u) \sum_v (f(u) - f(v))(\phi(v) P_{v,u} + \phi(u) P_{u,v})}{\sum_{u \in \mathcal{V}} 2 g(u)^2 \phi(u)} \tag{50}$$

$$\geq \frac{\sum_u g(u) \sum_v (g(u) - g(v))(\phi(v) P_{v,u} + \phi(u) P_{u,v})}{\sum_u 2 g(u)^2 \phi(u)}$$

We resort the vertices in $\mathcal{V}$ such that $f(v_1) \geq f(v_2) \geq \dots \geq f(v_{|\mathcal{V}|})$. Also, we can change the direction of $w$ so that $\sum_{f(u) < 0} \phi(u) \geq \frac{1}{2} \geq \sum_{f(u) \geq 0} \phi(u)$.

From the definition of $\Phi(\mathcal{H}) = \min_{\mathcal{S} \subseteq \mathcal{V}} \Phi(\mathcal{S}) = \frac{|\partial \mathcal{S}|}{\min(vol(\mathcal{S}), vol(\bar{\mathcal{S}}))}$, for every $i$ such that $f(v_i) \geq 0$,

$$\Phi(\mathcal{H}) \leq \frac{\sum_{k \leq i < l} \phi(v_k) P_{v_k, v_l}}{\sum_{j \leq i} \phi(v_j)} \iff \Phi(\mathcal{H}) \sum_{j \leq i} \phi(v_j) \leq \sum_{k \leq i < l} \phi(v_k) P_{v_k, v_l} \tag{51}$$

One can validate that

$$\sum_u g(u) \sum_v (g(u) - g(v))(\phi(v) P_{v,u} + \phi(u) P_{u,v}) = \sum_u \sum_v (g(u) - g(v))^2 \phi(v) P_{v,u} \tag{52}$$

Continue with Equation 50 with Theorem 3 in (Chung, 2005),

$$
\begin{aligned}
\lambda &\geq \frac{\sum_u \sum_v (g(u) - g(v))^2 \phi(v) P_{v,u}}{\sum_u 2 g(u)^2 \phi(u)} \\
&= \frac{(\sum_u \sum_v (g(u) - g(v))^2 \phi(v) P_{v,u})^2}{\sum_u 2 g(u)^2 \phi(u) (\sum_u \sum_v (g(u) - g(v))^2 \phi(v) P_{v,u})} \\
&\geq \frac{(\sum_u \sum_v |g(u)^2 - g(v)^2| \phi(v) P_{v,u})^2}{8(\sum_u g(u)^2 \phi(u))^2} \\
&= \frac{(\sum_k \sum_{l>k} (g(v_k)^2 - g(v_l)^2)(\phi(v_l) P(v_l, v_k) + \phi(v_k) P(v_k, v_l)))^2}{8(\sum_u g(u)^2 \phi(u))^2} \\
&= \frac{(\sum_k (g(v_k)^2 - g(v_{k+1})^2) \sum_{i \leq k < j} (\phi(v_i) P(v_i, v_j) + \phi(v_j) P(v_j, v_i)))^2}{8(\sum_u g(u)^2 \phi(u))^2}
\end{aligned}
\tag{53}
$$

Using Equation 51, continue with Equation 53,

$$
\begin{aligned}
\lambda &\geq \frac{(\sum_k (g(v_k)^2 - g(v_{k+1})^2) \sum_{i \leq k < j} (\phi(v_i) P(v_i, v_j) + \phi(v_j) P(v_j, v_i)))^2}{8(\sum_u g(u)^2 \phi(u))^2} \\
&\geq \frac{(\sum_k (g(v_k)^2 - g(v_{k+1})^2) 2 \Phi(\mathcal{H}) \sum_{i \leq k} \phi(v_i))^2}{8(\sum_u g(u)^2 \phi(u))^2} \\
&= \frac{\Phi(\mathcal{H})^2 (\sum_i \phi(v_i) \sum_{k \geq i} (g(v_k)^2 - g(v_{k+1})^2))^2}{2(\sum_u g(u)^2 \phi(u))^2} \\
&= \frac{\Phi(\mathcal{H})^2 (\sum_i \phi(v_i) g(v_i)^2)^2}{2(\sum_u g(u)^2 \phi(u))^2} \\
&= \frac{\Phi(\mathcal{H})^2}{2}
\end{aligned}
\tag{54}
$$

$\square$

# B   Algorithm Complexity

## B.1   Worst-case Time Complexity of HyperClus-G

The pseudo-code of HyperClus-G is given in Algorithm 1. Assume we have direct access to the support of each $\gamma_e$. Assume the number of hyperedge-vertex connections is $m$, which is the sum of the sizes of the support sets of all $\gamma_e$.

The computation of $R$ and $W$ takes $O(m)$. The constructed $R$ and $W$ both have $m$ non-zero entries. The construction of $D_\mathcal{V}$ takes $O(|\mathcal{V}|)$ and the construction of $D_\mathcal{E}$ takes $O(|\mathcal{E}|)$. Given that each hyperedge has at least 2 vertices and each vertex has at least one hyperedge incident to it, step 1 takes $O(m)$.

Given that $W$ and $R$ both have $m$ non-zero elements, the multiplication $P = D_\mathcal{V}^{-1} W D_\mathcal{E}^{-1} R$ takes $O(m^2)$ using CSR format sparse matrix. Therefore step 2 takes $O(m^2)$.

Computing the stationary distribution $\phi$ of $P$ takes $O(|\mathcal{V}|^2)$ using power iteration. Constructing the stationary distribution matrix $\Pi$ from $\phi$ takes $O(|\mathcal{V}|)$. Computing the random-walk-base hypergraph Laplacian takes $O(|\mathcal{V}|)$. Therefore step 3 and step 4 takes $O(|\mathcal{V}|^2)$.

Step 5, computing the eigenvector associated with the second smallest eigenvalue, takes $O(|\mathcal{V}|^3)$. And step 6 takes $O(|\mathcal{V}|)$. Therefore step 5 and step 6 together take $O(|\mathcal{V}|^3)$.

Therefore the worst-case complexity of HyperClus-G is

$$O(m) + O(m^2) + O(|\mathcal{V}|^2) + O(|\mathcal{V}|^3) \in O(m^2 + |\mathcal{V}|^3) \tag{55}$$

### B.2  Worst-case Space Complexity of HyperClus-G

We make the same assumption as in Section B.1. The storage of $R, W, D_\mathcal{V}, D_\mathcal{E}$ takes $O(m)$. The storage of $P$ and $L$ takes $O(|\mathcal{V}|^2)$. The storage of stationary distribution and the intermediate results takes $O(|\mathcal{V}|)$. The intermediate results for eigendecomposition take $O(|\mathcal{V}|^2)$.

Therefore, the worst-case space complexity for HyperClus-G is

$$O(m) + O(|\mathcal{V}|^2) + O(|\mathcal{V}|) + O(|\mathcal{V}|^2) \in O(m + |\mathcal{V}|^2) \tag{56}$$

In our experiments, the largest dataset Covertype's GPU usage is 6 Gigabytes.

## C  Experiemnt Details

### C.1  Environments

We run all our experiments on a Windows 11 machine with a 13th Gen Intel(R) Core(TM) i9-13900H CPU, 64GB RAM, and an NVIDIA RTX A4500 GPU. One can also run the code on a Linux machine. All the code of our algorithms is written in Python. The Python version in our environment is 3.11.4. In order to run our code, one has to install some other common libraries, including PyTorch, pandas, numpy, scipy, and ucimlrepo. Please refer to our README in the code directory for downloading instructions.

### C.2  Datasets

Table 10 summarizes the statistics and attributes of each dataset we used in the experiments.

**Mushroom** (`https://archive.ics.uci.edu/dataset/73/mushroom`) includes categorical descriptions of 8124 mushrooms in the Agaricus and Lepiota Family. Each species is labeled as edible or poisonous. This dataset contains missing data in the "stalk-root" feature. As (Zhou et al., 2006) did, we removed the feature that contains missing labels. The two clusters have 4208 and 3916 instances, respectively.

**Rice** (`https://archive.ics.uci.edu/dataset/545/rice+cammeo+and+osmancik`), a.k.a., Rice (Cammeo and Osmancik). A total of 3810 rice grain's images were taken for the two species, processed, and feature inferences were made. 7 morphological features were obtained for each grain of rice. This dataset does not have missing data. The feature types are real, including integer values, continuous values, and binary values. The two clusters have 2180 and 1630 instances, respectively. For each continuous feature, we first find the maximum value of this feature, then normalize all the numbers in this feature by the maximum value. This results in 10 bins of equal size $[0, 0.1], (0.1, 0.2], ...(0.9, 1]$. Then, we convert the quantified bins to categorical features by using a categorical feature to mark which bins a continuous value originally belongs to.

**Car** (`https://archive.ics.uci.edu/dataset/19/car+evaluation`), a.k.a., Car Evaluation. Originally, this dataset contained 1728 cars with labeled acceptability related to 6 features. This dataset does not have missing data. We extract all the cars that are labeled "good" or "vgood" (very good) and construct a smaller dataset of 134 cars. The two clusters have 65 and 69 instances, respectively.

**Digit-24** is a subset of **Digit** (`https://archive.ics.uci.edu/dataset/80/optical+recognition+of+handwritten+digits`), a.k.a. Optical Recognition of Handwritten Digits. This dataset contains a matrix

of 8x8 where each element is an integer in the range 0, 1, ..., 16. This reduces dimensionality and gives invariance to small distortions. This dataset does not have missing data. The original Digit datasets have 10 classes, and we extract all the instances of numbers 2 and 4 to construct Digit-24 for 2-way clustering. We simply regard the integer feature type to be categorical, that each integer is one category. The number of instances in each digit class is approximately the same.

**Covertype** (`https://archive.ics.uci.edu/dataset/31/covertype`). The task of this dataset is the classification of pixels into 7 forest cover types based on attributes such as elevation, aspect, slope, hillshade, soil-type, and more. We follow (Hein et al., 2013) and extract the instances of classes 4 and 5 to construct a dataset for 2-way clustering. The numerical feature values in this dataset can vary within a large range. Therefore, we first quantize the numerical values into 10 bins of equal size. We use the same strategy as described in the Rice dataset above to convert the features into categorical features. The two clusters have 9493 and 2747 instances, respectively.

**Zoo** (`https://archive.ics.uci.edu/dataset/111/zoo`) is a simple database containing 17 Boolean-valued attributes. We simply regard the integer feature type to be categorical, that each integer is one category. The seven clusters of instances have 41, 20, 13, 10, 8, 5 and 4 instances, respectively.

**Wine-567** (`https://archive.ics.uci.edu/dataset/186/wine+quality`), a.k.a., Wine Quality. The goal of this dataset is to model wine quality based on physicochemical tests. In the original dataset, each instance has a quality score between 0 to 10. The majority of instances have scores 5, 6, or 7. We extract all the instances that have scores 5, 6, or 7 to construct our Wine-567 dataset for 3-way clustering. We quantize the real feature values as described in the Rice dataset above, except that the number of bins is 20 instead of 10. The three clusters of instances have 2836, 2138, and 1079 instances, respectively.

**Letter** (`https://archive.ics.uci.edu/dataset/80/optical+recognition+of+handwritten+digits`)m a.k.a. Letter Recognition, which is a database of character image features, aiming to identify the letter. We extract all the instances of numbers 2 and 4 to construct our Letter dataset. The objective is to identify each of a large number of black-and-white rectangular pixel displays as one of C, I, L, or M. The number of instances in each letter class is approximately the same. We simply regard the integer feature type to be categorical, that each integer is one category.

Table 10: Statistics of Constructed Hypergraphs in Global Partitioning Experiments

| Hypergraph | $|\mathcal{V}|$ | $|\mathcal{E}|$ | $\sum_{v \in \mathcal{V}} E(v)$ | # classes | $|\mathcal{E}|$ clique | # original features | Subject Are | Feature Type |
|---|---|---|---|---|---|---|---|---|
| Mushroom | 8124 | 111 | 162480 | 2 | 65991252 | 22 | Biology | Categorical |
| Rice | 3810 | 3838 | 17410 | 2 | 14272406 | 7 | Biology | Real |
| Car | 134 | 16 | 804 | 2 | 23821 | 6 | Others | Categorical |
| Digit-24 | 1125 | 880 | 69750 | 2 | 69750 | 64 | Computer Science | Integer |
| Covertype | 12240 | 111 | 428400 | 2 | 149805360 | 52 | Biology | Categorical, Integer |
| Zoo | 101 | 36 | 1616 | 7 | 10100 | 16 | Biology | Categorical, Integer |
| Wine-567 | 6053 | 136 | 66583 | 3 | 36630116 | 11 | Business | Real |
| Letter | 3044 | 228 | 48704 | 4 | 8098180 | 16 | Computer Science | Integer |
| Digit | 5620 | 912 | 348440 | 10 | 31578780 | 64 | Computer Science | Integer |

Table 10 shows the statistics of our datasets used in the global partitioning experiments. The meanings of columns are the name of the hypergraphs, number of vertices/instances, number of hyperedges, number of hyperedge-vertex connections, number of classes, number of edges in the clique expansion graphs, number of original features in the dataset, the subject area of the dataset, and the types of features.

### C.3 Global Partitioning Metrics

The F1 score between two sets $\mathcal{A}_m$ and $\mathcal{A}_c$ is defined as

$$
\begin{aligned}
&TP\textit{(True Positive)} = |\mathcal{A}_m \cap \mathcal{A}_c| \\
&FP\textit{(False Positive)} = |\mathcal{A}_m \setminus \mathcal{A}_c| \\
&FN\textit{(False Negative)} = |\mathcal{A}_c \setminus \mathcal{A}_m| \\
&precision = \frac{TP}{TP + FP} \\
&recall = \frac{TP}{TP + FN} \\
&F1 = \frac{2 \times precision \times recall}{precision + recall}
\end{aligned}
\tag{57}
$$

Assume for $k$-way clustering (in this section, k can be 2), the algorithm returns the result $\mathcal{S}_1, \mathcal{S}_2, ..., \mathcal{S}_k$, then the NCut value is computed as

$$
NCut(\mathcal{S}_1, ..., \mathcal{S}_k) = \sum_{i=1}^{k} \frac{|\partial \mathcal{S}_i|}{vol(\mathcal{S}_i)} = \sum_{i=1}^{k} \frac{\sum_{u \in \mathcal{S}, v \in \bar{\mathcal{S}}} \phi(u) P_{u,v}}{vol(\mathcal{S})} \in [0, k]
\tag{58}
$$

Specifically, for 2-way NCut, we have another equivalent Definition 16. Assume the actual labeled classes are $\mathcal{V}_1, \mathcal{V}_2, ..., \mathcal{V}_k$, where $\mathcal{V}_i$ is all the vertices in class $i$. We first compute the F1 scores between $\mathcal{S}_i$ and $\mathcal{V}_j$ for $i, j \in \{1, 2, ..., k\}$, then greedily match the resulted sets with the actual labeled classes (Kollias et al., 2012). For example, if we have three classes with the F1 matrix, where $\mathbf{F}_{i-1,j-1}$ is the F1 score of $\mathcal{S}_i$ and $\mathcal{V}_j$

$$
\mathbf{F} = \begin{bmatrix} 0 & 0.9 & 0 \\ 0.8 & 0 & 0 \\ 0 & 0.7 & 0.6 \end{bmatrix}
\tag{59}
$$

Then, we first match $\mathcal{S}_1$ with $\mathcal{V}_2$ because they have the highest F1 score of 0.9. Then we match $\mathcal{S}_2$ with $\mathcal{V}_1$ because they have the second largest F1 score of 0.8. Then we try to match $\mathcal{S}_3$ with $\mathcal{V}_2$ because they have a third largest F1 score of 0.7, but $V_2$ has already been matched with $\mathcal{S}_3$, so we cannot match it again. Then, we try to match $\mathcal{S}_3$ with $\mathcal{V}_3$ and this time none of $\mathcal{S}_3$ and $\mathcal{V}_2$ has been matched. As all the sets have been matched, the greedy-match algorithm ends.

We calculate the weighted F1 score as the final metric of clustering. For each match $\mathcal{S}_{j_i}$ with $\mathcal{V}_i$, we weigh its F1 by number of instances in $\mathcal{V}_i$,

$$
Weighted\ F1 = \sum_{i=1}^{k} \frac{|\mathcal{V}_i|}{\sum_{i=1}^{k} |\mathcal{V}_i|} \mathbf{F}_{j_i-1,i-1} \in [0, 1]
\tag{60}
$$

### C.4 From 2-way Clustering to $k$-way Clustering

We have two strategies from 2-way clustering to $k$-way clustering. For the datasets that each actual labeled class has a similar number of instances, we apply 2-way clustering on the largest cluster every time to split it into two clusters. For $k$-way clustering, this process calls the 2-way clustering $k$-1 times.

When the number of instances in each actual labeled class varies a lot, when we have $l$ clusters, we call 2-way clustering on each cluster to get $l + 1$ clusters, then pick the best of $l$ results in terms of NCut. For $k$-way clustering, this process calls the 2-way clustering $\frac{k(k-1)}{2}$ times.

Another promising future direction is to develop a direct spectral analysis for the k-way case. This, however, involves non-trivial technical challenges and lies beyond the scope of the present work. We note that such an extension should also draw inspiration from analogous results in graphs, such as the k-way NCut formulation.

**Definition 25** (k-way Normalized Cut)**.** *Let $H = (V, E, \omega, \gamma)$ be an EDVW hypergraph with stationary distribution $\phi$ and transition matrix $P$. For any k-way partition $\{S_1, \ldots, S_k\}$ of $V$ into non-empty disjoint subsets, the k-way normalized cut (NCut) is defined as*

$$NCut(S_1, \ldots, S_k) \; = \; \sum_{i=1}^{k} \frac{|\partial S_i|}{vol(S_i)} \; = \; \sum_{i=1}^{k} \frac{\sum\limits_{u \in S_i, \, v \in \overline{S_i}} \phi(u) P_{uv}}{\sum\limits_{u \in S_i} \phi(u)}, \tag{61}$$

*where $vol(S_i) = \sum_{u \in S_i} \phi(u)$ is the stationary volume of cluster $S_i$, and $|\partial S_i| = \sum_{u \in S_i, \, v \in \overline{S_i}} \phi(u) P_{uv}$ denotes the boundary volume between $S_i$ and its complement.*

### C.5 Baselines

**CLIQUE++.** For each hyperedge $e$, each $u, v \in e$ with $u \neq v$, we add an edge $uv$ of weight $w(e)$. Then, we compute the adjacency matrix $A$ and degree matrix $D$. We calculate the graph Laplacian matrix by $L = D - A$. Finally, we do eigen-decomposition for the random walk Laplacian $L_{RW} = D^{-1}L$, whose eigenvalues are associated with the graph NCut value (Hamilton, 2020). We calculate the eigenvector associated with the second smallest eigenvalue for global partitioning: we put the non-negative entries as one cluster and the negative entries as another.

**STAR++.** For each hyperedge $e$, we introduce a new vertex $v_e$. For each vertex $u \in e$, we add an edge $uv_e$ of weight $w(e)/|e|$. After converting the hypergraph into a star graph, we do the same algorithm for global partitioning as in CLIQUE++.

**DiffEq** (Takai et al., 2020). We directly use the official code[4] of this algorithm. This method sweeps over the sweep sets obtained by differential equations for local clustering. For global partitioning, it simply calls local clustering for every vertex and returns the best in terms of conductance. Originally, this algorithm could only take one starting vertex for local clustering. We modified the code to add one additional vertex that connects the 5 starting vertices in each observation. Then regard the newly added vertex as the starting vertex, so that it can obtain the local cluster for the given 5 starting vertices.

**node2vec** (Grover & Leskovec, 2016). node2vec embeds a graph using random walks. Since we have the random walk matrix $P$ on the hypergraph, we construct a directed weighted graph by $P$ as the input of node2vec. However, given that there are too many non-zero entries in $P$, node2vec is extremely slow. Therefore, after we get $P$, we only keep the entries of $P$ that are larger than a small threshold. This threshold is tuned so that the (1) execution time is acceptable; (2) the NCut value and F1 value of the result are both near convergence. We did not modify other default hyperparameters in node2vec. After we obtain the vertex embedding, we call KMeans from scikit-learn to directly obtain k clusters for $k$-way clustering.

**event2vec/hyperedge2vec** (Fu et al., 2019). We directly use the official code[5] of this algorithm. event2vec was originally designed for heterogeneous graphs. For example, in a citation network that has publications and authors, each publication is an event. We notice that here an event is equivalent to a hyperedge. We first convert the hypergraph into a STAR graph, then we regard each added vertex as an event and call event2vec to obtain vertex embeddings. We tune the training epochs near default so that (1) execution time is acceptable; (2) the NCut value and F1 value of the result are both near convergence. We did not modify other default hyperparameters in event2vec. After we obtain the vertex embedding, we call KMeans from scikit-learn to directly obtain k clusters for $k$-way clustering.

---

[4]`https://github.com/atsushi-miyauchi/Hypergraph_clustering_based_on_PageRank`
[5]`https://github.com/guoji-fu/Event2vec`

### C.6   Supplementary Experiment Data

### C.6.1   Standard Deviations of Nondeterministic Algorithms on Global Partitioning

We report the standard deviation of nondeterministic algorithms on global partitioning. The nondeterministic algorithms are node2vec + kmeans and hyperedge2vec + kmeans. The standard deviations of the NCut are reported in Table 11 and those of the F1 scores are reported in Table 12.

Table 11: Standard Deviations of NCut on Global Partitioning Task.

| Method | 2-way Clustering | | | | | $k$-way Clustering (k $\geq$ 3) | | | |
|---|---|---|---|---|---|---|---|---|---|
| | Mushroom | Rice | Car | Digit-24 | Covertype | Zoo | Wine | Letter | Digit |
| node2vec | 1e-5 | 0.049 | 0.006 | 1e-4 | 1e-5 | 0.014 | 0.001 | 0.046 | 0.039 |
| hyperedge2vec | 0.002 | 0.038 | 0.006 | 0.001 | 0.024 | 0.008 | 0.026 | 0.017 | 0.009 |

Table 12: Standard Deviations of F1 scores on Global Partitioning Task

| Method | 2-way Clustering | | | | | $k$-way Clustering (k $\geq$ 3) | | | |
|---|---|---|---|---|---|---|---|---|---|
| | Mushroom | Rice | Car | Digit-24 | Covertype | Zoo | Wine | Letter | Digit |
| node2vec | 1e-5, 1e-5 | 0.056, 0.054 | 0.032, 0.031 | 7e-4, 7e-4 | 0.027, 0.024 | 0.023 | 0.001 | 0.078 | 0.079 |
| hyperedge2vec | 0.196, 0.197 | 0.110, 0.109 | 0.174, 0.171 | 0.050, 0.050 | 0.124, 0.129 | 0.046 | 0.010 | 0.060 | 0.006 |

### C.6.2   Time Comparison on Global Partitioning

The execution time of all the methods is reported in Table 13. Our HyperClus-G outperforms the baseline methods on 8/9 datasets. Note that DiffEq is written in C#, while others are written in Python. It may not be a fair comparison since C# is one of the fastest programming languages, but we still report the total execution time (including compiling) of all the algorithms for reference.

After STAR++ expansion, the dimension of the matrix gets larger because we need to introduce additional vertices into the graph. For CLIQUE++ expansion, since we convert the hyperedge to the fully connected CLIQUE graph, the conversion itself makes the program slower. Also, the random walks and calculation of Laplacian will be slower compared to HyperClus-G. DiffEq finds the global partition by actually finding a local cluster and taking its complementary set. As the graph becomes large, sweeping over to find the local cluster will consume more time. node2vec and event2vec/hyperedge2vec need to train the embedding model, which consumes much time. We have tuned the number of training epochs to let the model stop near convergence.

Table 13: Execution Time Comparison($\downarrow$) on Global Partitioning Task (unit: seconds).

| Method | 2-way Clustering | | | | | $k$-way Clustering (k $\geq$ 3) | | | |
|---|---|---|---|---|---|---|---|---|---|
| | Mushroom | Rice | Car | Digit-24 | Covertype | Zoo | Wine | Letter | Digit |
| STAR++ expansion | 72.82 | 50.80 | 1.43 | 3.37 | 318.55 | 0.654 | 32.18 | 37.54 | 59.05 |
| CLIQUE++ expansion | 216.04 | 13.32 | 1.38 | 7.50 | 1099.03 | 0.629 | 77.75 | 50.25 | 297.37 |
| DiffEq (C#) | 39.03 | 8.15 | 1.43 | 18.36 | 149.49 | 8.79 | 33.09 | **20.94** | 613.22 |
| node2vec + kmeans | 101.05 | 73.08 | 3.54 | 12.63 | 175.58 | 2.238 | 39.66 | 32.68 | 48.59 |
| hyperedge2vec + kmeans | 88.96 | 47.03 | 5.59 | 20.29 | 367.53 | 4.62 | 65.68 | 28.40 | 113.53 |
| HyperClus-G(Ours) | **17.73** | **5.04** | **1.37** | **2.18** | **87.72** | **0.595** | **11.48** | 35.13 | **26.49** |
| $\frac{\text{HyperClus-G}}{\text{Best of Python Baselines}}$ ratio | 24.34% | 37.83% | 99.27% | 64.69% | 49.96% | 94.59% | 35.67% | 123.7% | 44.86% |

