# OpenReview forum: "Hypergraphs as Weighted Directed Self-Looped Graphs: Spectral Properties, Clustering, Cheeger Inequality"
_TMLR — Accepted by TMLR_

### Review · Reviewer_AWKa · 2025-04-13

**Summary Of Contributions:**

In the paper, the authors propose a novel approach to optimizes the Normalized Cut (NCut) value in hypergraphs with edge-dependent vertex weights (EDW) through a spectral clustering approach grounded in hypergraph Laplacian theory. The algorithm is called HyperClus-G, and approximates the optimal NCut value, which is the second-smallest eigenvalue, after building the random-walk-based Laplacian of
the hypergraph.

This approach seem to bring important advancements in the field of spectral hypergraphs analysis.

**Audience:**

Yes

**Broader Impact Concerns:**

I have no concerns about the ethical implications of the work

**Claims And Evidence:**

Yes

**Requested Changes:**

My only concern is about the structure of the paper. Starting from the abstract, which focuses more on what is missing on the literature than what the paper brings to the table, to the flow of the paper.

In particular, many concepts are introduced multiple times, and in combination with the heavy density of maths formulas, I found the paper hard to navigate. For example, the Main Results sections could be integrated in the Introduction one in a less formal way, removing or compressing some maths formulation. Additionally, some sentences seem disconnected from the others, and could be better integrated.

However, such are minor problems, and even if such minor changes could improve the readability, I suggest accepting the manuscript.

**Strengths And Weaknesses:**

To the best of my knowledge, the proposed approach, called HyperClus-G, is truly novel and brings a lot to the table of spectral hypergraphs analysis. The proposed approach incorporates  EDVW info instead of losing them by approximation, leading to stronger results, in particular:

1. Directly incorporates edge-dependent weights, as opposed to other, which approximates hypergraphs as graphs, losing EDVW info
2. Matches or exceeds F1 scores, while other approaches reach suboptimal due to information loss in conversion
3. HyperClus-G runs up to 10× faster than STAR++/CLIQUE++ on datasets like Rice and Covertype, despite handling more complex hypergraph structures
4. Experiments show the second smallest eigenvalue of closely approximates the NCut value achieved by HyperClus-G, with relative errors under 25% (suggesting further room for improvements)

Additionally, such results are supported by theoretical proofs:

1. HyperClus-G guarantees that the NCut of its partitions satisfies NCut≤O(NCut*) , where NCut* is the optimal value
2. he algorithm’s performance is bounded by the hypergraph Cheeger inequality, linking λ2 to the conductance of the partition

However, the proposed approach has its limitations. In particular, storing the matrices can be prohibitive and computing the second eigenvector could be a bottleneck due to the complexity required for the exact solution; the employed Recursive Bipartitioning has its own drawbacks (such as accumulates errors in hierarchical divisions or suboptimally results on datasets with imbalanced cluster densities). Said so, the method is strong and convincing.

---

> ### Author Response · Authors · 2025-09-08
>
> We sincerely thank the reviewer for their recognition of the paper’s contributions and constructive feedback. We have incorporated these suggestions into the revision to improve both structure and readability. The key changes made in response to your suggestions are highlighted in orange.
>
> > Computation.
>
> We agree that storing large matrices and computing exact solutions can be challenging for large hypergraphs, as this is a common bottleneck for hypergraph algorithms. In this work, we analyzed both the space and time complexity in Appendix B, and empirically validated that our method can handle hypergraphs with up to 10,000 nodes within practical time and GPU memory limits.
>
>
> > Structure and readability.
>
> We acknowledge that the density of mathematical formulations made the paper challenging to navigate. To address this, we made the following changes.
>
> **Structural clarity.** We added Table 2, which concisely summarizes our developed formulations and highlights both what was missing in prior work and what this paper contributes.
>
> **Improved readability.** We introduced Figure 2, which illustrates the logical flow of the paper by explicitly mapping which theorems build upon which definitions.
>
> **Conciseness.** We reduced redundancy and verbosity to ensure that the paper is more compact while retaining all key content.
>
> We hope these improvements make the paper more accessible and easier to follow. Should you have any further questions, we are more than happy to discuss.

---

### Review · Reviewer_zoDJ · 2025-08-24

**Summary Of Contributions:**

This paper presents spectral properties, clustering algorithms, and the Cheeger inequality for edge-dependent vertex weights (EDVW) modeling of hypergraphs. It establishes an algebraic connection between the combinatorial Normalized Cut optimization objective and the spectral Rayleigh Quotient of a random-walk-based hypergraph Laplacian. Based on this connection, it proposes `HyperClus-G`, a spectral clustering algorithm that is claimed to achieve approximately linear optimality for partitioning EDVW hypergraphs.

**Audience:**

Yes

**Broader Impact Concerns:**

There is no issue since this is a theoretical paper.

**Claims And Evidence:**

Yes

**Requested Changes:**

- (p2) I recommend introducing terminology before introducing the main results.
- (p3) The authors claimed that:

    > This is the first work regarding the Rayleigh Quotient on hypergraphs.

    The reviewer would like to point out that (Jost & Mulas, 2019; Mulas et al., 2022) also proposed the Rayleigh Quotient on hypergraphs. Although they assumed the EIVW model, the authors need to clarify that the proposed work is the first work on the EDVW hypergraphs model.

- When introducing spectral properties in Section 3, there is a need to explain what stationary distribution will be used. In the case of a connected hypergraph assumed in the main text, it is irreducible and aperiodic, so by the Perron-Frobenius theorem, there exists a unique stationary distribution with all positive entries (Section A.4). I believe mentioning this part would ensure readers understand that concepts introduced in section 3 such as volume, Rayleigh quotient, and normalized cut are well-defined.
- (p7) Since Theorem 17 is identical to Theorem 1, it would be better for readability not to define the labeling anew.
- (p7) In Eq.(18), the referenced Eq.(42) is identical to Eq.(16), so I recommend making the labeling consistent to avoid confusion for readers.
- (p9) Theorem 17 is identical to Theorem 1.
- (p9) Using the “$\leq$” → Using the RHS.
- (p9) Lemma 22 appears to have more of a corollary character than a lemma.
- (p18) In Section A.6, the calculations in Eq.(39), (40), 41) are somewhat redundant. There is insufficient explanation for non-trivial propositions, and it is unnecessary to write out trivial calculations at length.
- (p18) In Eq.(42), the numerator and denominator are switched for $u \in \bar{\mathcal{S}}$.
- (p18) In Eq.(43), $X(u), X(v) \rightarrow x(u), x(v)$.
- (p19) The proof in Section A.8 is also verbose. According to Definition 12, $ |\partial S| \leq \text{vol}(S) $ and $ |\partial \bar{S}| \leq \text{vol}(\bar{S}) $ hold, so it is trivial.
- (p19) Change "\leq" → RHS, "\geq" → LHS.

[Chitra & Raphael, 2019] Random walks on hypergraphs with edge-dependent vertex weights. ICML, 2019.

[Jost & Mulas, 2019] Hypergraph Laplace operators for chemical reaction networks. Advances in Mathematics, 2019.

[Mulas et al., 2022] Graphs, Simplicial Complexes and Hypergraphs: Spectral Theory and Topology. Higher-Order Systems, 2022.

**Strengths And Weaknesses:**

Strengths

- The theoretical core of the paper is strong, well-motivated, and impactful. The strength of the theoretical results presented in this paper is that they establish spectral techniques that can be used in the EDVW model proposed by (Chitra & Raphael, 2019).
- Based on the EDVW hypergraph random walk theory, the paper presents an algebraic connection between Rayleigh quotient and Normalized Cut (Theorem 1), and proves the approximately linear optimality of the proposed hypergraph spectral clustering algorithm (Theorem 2). These results appear to be a good extension of spectral graph theory to EDVW hypergraphs.

Weaknesses

- A significant number of theoretical proofs need revision. Some proofs show elementary mistakes in formal logic, and there are unnecessarily verbose sections that require effort to enhance the expertise. The labeling of theorems is also cumbersome as one needs to move back and forth multiple times between the main text and the appendix while reading. Efforts are needed to improve the readability of the paper.
- The proof of the Cheeger inequality presented in the paper borrows proof techniques from (F. Chung, 2005). This proof appears to tediously apply calculations used in the proof of the Cheeger inequality for weighted directed graphs corresponding to hypergraphs, as mentioned in (Chitra & Raphael, 2019) (Appendix A.9). In the reviewer's opinion, the authors should be more cautious about claiming that this aspect was "non-proved" in previous papers or that they have "upgraded" existing results. The reviewer recommends that the authors clearly articulate the specific differences between the Cheeger inequality proposed in this paper and the results in existing papers, and illuminate their relationship, which would be a better way to explain their contribution.

[Chitra & Raphael, 2019] Random walks on hypergraphs with edge-dependent vertex weights. ICML, 2019.

[F. Chung, 2005] Laplacians and the Cheeger inequality for directed graphs, Annals of Combinatorics, 9:1-19, 2005.

---

> ### Author Response · Authors · 2025-09-08
>
> We sincerely thank the reviewer for their recognition of the paper’s contributions and constructive feedback. We incorporate their feedback into the manuscript during revision and hope to address their concerns. The key changes made in response to your suggestions are highlighted in deep green.
>
> > Weakness 1: Revision of theories.
>
> We greatly appreciate the careful review and the identification of minor errors that had escaped our attention. We revise the paper to address these issues point by point. In addition, to improve readability and help readers navigate the paper, we have added Figure 2, which illustrates the logical flow by showing how theorems depend on the definitions.
>
> > Weakness 2: how this paper’s Cheeger Inequality differs and improves over previous work.
>
> We add Table 2 to present a clearer comparison to previous work. In particular, earlier studies did not provide a well-defined conductance, and did not provide a complete proof of the Cheeger Inequality. As a result, they relied on the wrong eigenvalue. Specifically, (Chitra & Raphael 2019) formulated the Cheeger Inequality using the combinatorial Laplacian L, whereas the correct formulation requires the symmetric normalized Laplacian L_sym​, which has a different spectrum. Our work resolves this issue by presenting the first complete proof of the Cheeger Inequality for EDVW hypergraphs, using some results from (Chung 2005).
>
> To be more precise and balanced, we also revised our wording regarding prior work and removed expressions such as “non-proved results” or “conjecture.”
>
> > Requested Change 1: making the main results easier to read.
>
> We agree that introducing terminology before the main results could improve readability. However, moving all formal definitions into the introduction would risk overwhelming the reader and obscuring the paper’s core contributions. To this end, we added Table 2, which concisely summarizes our key formulations and serves as a handy reference when reading the main results. We hope this strikes a balance between accessibility and conciseness.
>
> > Requested Change 2: Overclaim of Cheeger Inequality on Hypergraphs.
>
> We thank the reviewer for pointing out the relevant references on Cheeger Inequality in the EIVW hypergraph setting. In the revised paper (page 3), we have added a discussion to clarify this distinction when making our claim.
>
> > Requested Change 3: Clarify the stationary distribution when introducing it.
>
> In the revised paper (page 5), we have added clarification on both the significance and the guaranteed existence of the stationary distribution to ensure that subsequent concepts such as volume, Rayleigh quotient, and NCut are well-defined.
>
>
> > Requested Changes 4, 5, 6: Labeling of theorems and equations.
>
> We have corrected the labeling issues in the revised paper. Specifically, Theorem 17 has been merged with Theorem 1 since they are identical, and the reference to Eq. (42) has been updated to Eq. (16).
>
>
> > Requested Change 7, 13: use LHS and RHS instead of “\geq” and “\leq”.
>
> This has been fixed in the revised paper (page 9, 20). We appreciate the reviewer for pointing it out.
>
> > Requested Change 8: change Lemma to Corollary.
>
> This has been fixed in the revised paper (page 9). We appreciate the reviewer for pointing it out.
>
>
> > Requested Change 9, 12: revise proofs.
>
> For Requested Change 9, we have reduced verbosity in the derivations of Eq. (39) and Eq. (40). For Eq. (41) (now Eq. (42) on page 18), we added an explicit equation to clarify the non-trivial step and removed redundant calculations to streamline the proof.
>
> For Change 12, following the suggestion, we simplified the proof in Appendix A.8, as reflected in the revised paper (page 19).
>
>
>
> > Requested Change 10, 11: Typos.
>
> These typos have been fixed in the revised paper (page 18, 19). We appreciate the reviewer for pointing them out.
>
>
> We hope our efforts can further improve this paper and address your concerns. Should you have any further questions, we are more than happy to discuss.

---

> > ### Comment · Reviewer_zoDJ · 2025-09-08
> >
> > Upon reviewing the revised version of the manuscript, I confirm that the authors have comprehensively addressed all concerns raised. I appreciate authors’ work.
> >
> > Based on these improvements, I would like to recommend this paper for acceptance to TMLR.

---

### Review · Reviewer_K9i9 · 2025-08-26

**Summary Of Contributions:**

This paper focuses on developing fundamental theory for hypergraphs, with a specific focus on the edge-dependent vertex weights (EDVW) model proposed in prior work. Specifically, hypergraphs are useful in modeling higher-order relations between groups and recent work has aimed to formalize hypergraphs that can capture such structure and design algorithms to learn them. However, as this work is nascent, this area lacks spectral theory akin to classical graph-based learning techniques. The present work aims to fill this gap by developing and analyzing the following notions for the EDVW hypergraph model:
- Spectral theoretical notions, such as Rayleigh quotient, NCut, volume, and conductance
- A clustering algorithm that is approximately linearly optimal for both NCut and conductance
- A Cheeger inequality for EDVW hypergraphs
- Empirics validating their approach, showing that their clustering algorithm improves performance.

**Audience:**

Yes

**Claims And Evidence:**

Yes

**Requested Changes:**

**Discussion with some results in (Chitra & Raphael, 2019)**

It would be beneficial to get a better sense of how some of the results in this work are different or improve upon those in prior work. For example, Theorem 5.1 in (Chitra & Raphael, 2019) also provides a bound on the least eigenvalue of the Laplacian while this work focuses on the second smallest eigenvector of the normalized laplacian. It would be good to more explicitly state the benefits of the new Theorem 3 and what are the advances potentially in terms of the proof technique, since both Theorem 5.1 and Theorem 3 use results from (Chung, 2005) to prove it. This discussion could also arise while defining the new Rayleigh quotient near Theorem 15.

**Minor changes**
- For the proof of Lemma 22, it would be better to make the $\approx$ in (27) more rigorous by perhaps referencing/showing how NCut is big O of $\lambda$
- Some sections have no narrative text between Definitions, Assumptions, or results (e.g., Definition 6, 7 -> Theorem 8). It could help to add narrative text between these for improved readability.

**Strengths And Weaknesses:**

**Strengths**
- To the reviewer's knowledge, this work presents some of the first results on spectral theory for EDVW hypergraphs.
- The paper is well-motivated.
- Empirically the proposed clustering algorithm (HyperClus-G) consistently performs well across several datasets and is on-par or outperforms other relevant methods.

**Weaknesses**
- The theory is limited to the $2$-way global partitioning case. It could be interesting to extend the theory to the case of $k > 2$-way global partitioning.
- I think that there are certain aspects of the presentation that can be improved. For example, I think that the paper could benefit from a more in-depth discussion of how the current work improves upon prior work and what some of the main differences are in the results. This is discussed more in the requested changes.

---

> ### Author Response · Authors · 2025-09-08
>
> We sincerely thank the reviewer for their recognition of the paper’s contributions and constructive feedback. We incorporate their feedback into the manuscript during revision and hope to address their concerns. The key changes made in response to your suggestions are highlighted in blue.
>
>
> > Weakness 1: extending to k-way (k>2) global partitioning.
>
>
> We acknowledge the importance of generalizing to k-way partitioning, which motivated us to include k-way experiments using recursive 2-way partitioning. However, extending our 2-way formulations and theoretical properties to the k-way setting is non-trivial and better left for future work. To provide some guidance, we include preliminary intuitions for such an extension in Appendix C.4 (page 24 and 25).
>
> > Weakness 2 & Requested Change 1: how our results improve from previous works
>
> We add Table 2, which summarizes our developed formulations, with direct comparisons to graphs and previous work. In particular, earlier studies did not provide a well-defined conductance, and did not provide a complete proof of the Cheeger Inequality. As a result, they relied on the wrong eigenvalue. Specifically, they formulated the Cheeger Inequality using the combinatorial Laplacian L, whereas the correct formulation requires the symmetric normalized Laplacian L_sym​, which has a different spectrum. Our work resolves this issue by presenting the first complete proof of the Cheeger Inequality for EDVW hypergraphs.
>
> > Minor 1: more rigorous \approx in Section 4.2.
>
> We have added a discussion in Section 4.2 (highlighted in blue). In brief, we show that \lambda can be bounded in terms of NCut up to a big-O factor, but the reverse bound is impossible, as counterexamples exist.
>
> > Minor 2: Readability.
>
> We have added more explanatory narrative (see colored text) and introduced Figure 2 to help readers navigate the formulations.
>
>
> We hope our efforts can further improve this paper and address your concerns. Should you have any further questions, we are more than happy to discuss.

---

### Author Response · Authors · 2025-09-08
**We thank the reviewers for their thoughtful feedback and upload a revision**

Dear Reviewers of Submission 4436,

We sincerely thank you for the detailed and constructive feedback, as well as the opportunity to further strengthen our paper. We are grateful for the recognition of this work’s motivation and technical contributions.

Across the reviews, several common concerns emerged:
- **Readability** of the paper. It contains dense mathematical formulations and is challenging to navigate.
- **Positioning relative to prior work**. In particular, clarifying how our contributions differ from and improve upon (Chitra & Raphael, 2019).
- Minor changes from each reviewer.


To address these concerns, we have substantially revised the manuscript (uploaded to the OpenReview portal). Changes are highlighted with colors (blue, deep green, orange for reviewer-specific requests, and brown for general improvements). In particular:
- We added Table 2 (on page 2), which summarizes our developed formulations, directly comparing them to graph formulations and to (Chitra & Raphael, 2019). We hope this could (1) provide a concise overview before readers encounter the main results, (2) clarify our technical advances and their relation to prior work, and (3) demonstrate how our formulations remain consistent with those for graphs.
- We introduced Figure 2 (on page 4), which illustrates the logical flow of the paper by mapping which theorems depend on which definitions. We hope this will help readers navigate the mathematical developments.
- We reduce verbosity to ensure the main content fits within 12 pages, address all minor issues, and will respond to each reviewer individually.

We hope these revisions improve both the readability and the clarity of our contributions, and we are happy to engage further on any remaining questions.


Once again, we sincerely thank the reviewers for their thoughtful feedback.


Sincerely,
Submission 4436 Authors

---

### Decision · Action_Editor_9ZKg · 2025-10-26

**Recommendation:** Accept as is

**Audience:**

Yes

**Audience Explanation:**

EDVW hypergraph models are more expressive and better fit to real-world problems, compared to EIVW hypergraph models. The HyperClus-G is a principled spectral clustering algorithm designed to operate on EDVW hypergraph models. While its practicability is limited, it might be of interest for those who are working on hypergraph models.

**Claims And Evidence:**

Yes

**Claims Explanation:**

The paper addresses a random-walk-based spectral theory for EDVW (edge-dependent vertex weights) hypergraphs. Algebraic connections among hypergraph NCut, Rayleigh Quotient, and graph Laplacians are well studied in Theorem 1.  Inspired by this algebraic connection, the paper presents HyperClus-G algorithm as a new spectral clustering algorithm designed to operate on EDVW hypergraphs. Its approximately linear optimality is shown in Theorem 2. A proof for the hypergraph Cheeger inequality is given in Theorem 3. Experiments demonstrate that the HyperClus-G outperforms baseline methods, although empirical studies are quite limited.